# WHEN DOES BIAS TRANSFER IN TRANSFER LEARNING?

## ABSTRACT

Using transfer learning to adapt a pre-trained "source model" to a downstream "target task" can dramatically increase performance with seemingly no downside. In this work, we demonstrate that there can exist a downside after all: bias transfer, or the tendency for biases of the source model to persist even after adapting the model to the target dataset. Through a combination of synthetic and natural experiments, we show that bias transfer both (a) arises in realistic settings (such as when pre-training on ImageNet or other standard datasets) and (b) can occur even when the target dataset is explicitly *de*-biased. As transfer-learned models are increasingly deployed in the real world, our work highlights the importance of understanding the limitations of pre-trained source models.

## 1 INTRODUCTION

Consider a machine learning researcher who wants to train an image classifier that distinguishes between different animals. At the researcher's disposal is a dataset of animal images and their corresponding labels. Being a diligent scientist, the researcher combs through the dataset to eliminate relevant spurious correlations (e.g., background-label correlations (Zhu et al., 2017; Xiao et al., 2020)), and to ensure that the dataset contains enough samples from all relevant subgroups.

Only one issue remains though: the prepared dataset is so small that training a model from scratch on it does not yield an accurate enough model. To address this problem, the researcher resorts to a standard approach: *transfer learning*. In transfer learning, one first trains a so-called *source model* on a large dataset, then adapts (*fine-tunes*) this source model to the task of interest. This strategy indeed often yields models that are far more performant.

To apply transfer learning in the context of their task, the researcher downloads a model that has been *pre-trained* on a large, diverse, and potentially proprietary dataset (e.g., JFT-300 (Sun et al., 2017) or Instagram-1B (Mahajan et al., 2018)). Unfortunately, such pre-trained models are known to have a variety of biases: for example, they can disproportionately rely on texture (Geirhos et al., 2019), or on object location/orientation (Barbu et al., 2019; Xiao et al., 2020; Leclerc et al., 2021). Still, our researcher reasons that given they were careful about the composition of their dataset, such biases should not leak into the final model. But is this really the case? More specifically,

*Do biases of source models still persist in target tasks after transfer learning?*

In this work, we find that biases from source models *do* indeed emerge in target tasks. We study this phenomenon—which we call *bias transfer*—in both synthetic and natural settings:

1. **Bias transfer through synthetic datasets.** We first use *backdoor attacks* (Gu et al., 2017) as a testbed for studying synthetic bias transfer, and characterize the impact of the training routine, source dataset, and target dataset on the extent of bias transfer. Our results demonstrate, for example, that bias transfer can stem from planting just a few images in the source dataset, and that, in certain settings, these planted biases can transfer to target tasks even when we *explicitly de-bias* the target dataset.

2. **Bias transfer via naturally-occurring features.** Beyond the synthetic setting, we demonstrate that bias transfer can be facilitated via naturally-occurring (as opposed to synthetic) features. Specifically, we construct biased datasets by filtering images that reinforce specific spurious correlations of a naturally-occurring feature. (For example, a dependence on

gender when predicting age for CelebA) We then show that even on target datasets that do not support this correlation, models pre-trained on a biased source dataset are still overly sensitive to that correlating feature.

3. **Naturally-occuring bias transfer.** Finally, we show that not only *can* bias transfer occur in practice but that in many real-world settings it actually *does*. Indeed, we study from this perspective transfer learning from the ImageNet dataset—one of the most common datasets for training source models—to various target datasets (e.g., CIFAR-10). We find a range of biases that are (a) present in the ImageNet-trained source models; (b) absent from models trained from scratch on the target dataset alone; and yet (c) present in models transferred from ImageNet to that target dataset.

## 2  BIASES CAN TRANSFER

Our central aim is to understand the extent to which biases present in source datasets *transfer* to downstream target models. Here, we define bias as a feature that a model relies on but is not causally linked to the target task (see Appendix A.3 for a more formal definition). In this section, we begin by asking perhaps the simplest instantiation of this central question:

*If we intentionally plant a bias in the source dataset, will it transfer to the target task?*

**Motivating linear regression example.**    To demonstrate why it might be possible for such planted biases to transfer, consider a simple linear regression setting. Suppose we have a large source dataset of inputs and corresponding (binary) labels, and that we use the source dataset to estimate the parameters of a linear classifier $\boldsymbol{w}_{src}$ with, for example, logistic regression. In this setting, we can define a *bias* of the source model $\boldsymbol{w}_{src}$ as a direction $\boldsymbol{v}$ in input space that the classifier is highly sensitive to, i.e., a direction such that $|\boldsymbol{w}_{src}^{\top}\boldsymbol{v}|$ is large.

Now, suppose we adapt (fine-tune) this source model to a target task using a target dataset of input-label pairs $\{(\boldsymbol{x}_i, y_i)\}_{i=1}^{n}$. As is common in transfer learning settings, we assume that we have a relatively small target dataset—in particular, that $n < d$, where $d$ is the dimensionality of the inputs $\boldsymbol{x}_i$. We then adapt the source model $\boldsymbol{w}_{src}$ to the target dataset by running stochastic gradient descent (SGD) to minimize logistic loss on the target dataset, using $\boldsymbol{w}_{src}$ as initialization.

With this setup, transfer learning will preserve $\boldsymbol{w}_{src}$ in all directions orthogonal to the span of the $\boldsymbol{x}_i$. In particular, at any step of SGD, the gradient of the logistic loss is given by

$$\nabla \ell_{\boldsymbol{w}}(\boldsymbol{x}_i, y_i) = (\sigma(\boldsymbol{w}^{\top}\boldsymbol{x}_i) - y_i) \cdot \boldsymbol{x}_i,$$

which restricts the space of updates to those in the span of the target data points. Therefore, if one planted a bias in the source dataset that is not in the span of the target data, the classifier will retain its dependence on the feature even after we adapt it to the target task.

**Connection to backdoor attacks.**    Building on our motivating example above, one way to plant such a bias would be to find a direction $\boldsymbol{u}$ that is orthogonal to the target dataset, add $\boldsymbol{u}$ to a subset of the *source* training inputs, and change the corresponding labels to introduce a correlation between $\boldsymbol{u}$ and the labels. It is worth noting that this idea bears a striking similarity to that of *backdoor attacks* (Gu et al., 2017), wherein an attacker adds a fixed "trigger" pattern (e.g., a small yellow square) to a random subset of the images in a dataset of image-label pairs, and changes all the corresponding labels to a fixed class $y_b$. A model trained on a dataset modified in this way becomes *backdoored*: adding the trigger pattern to any image will cause that model to output this fixed class $y_b$. Indeed, Gu et al. (2017) find that, if one adds a trigger that is absent from the target task to the source dataset, the final target model is still highly sensitive to the trigger pattern.

Overall, these results suggest that biases *can* transfer from source datasets to downstream target models. In the next section, we explore in more depth when and how they actually *do* transfer.

## 3  EXPLORING THE LANDSCAPE OF BIAS TRANSFER

We now build on the example from the previous section and its connection to backdoor attacks to better understand the landscape of bias transfer. Specifically, the backdoor attack framework enables

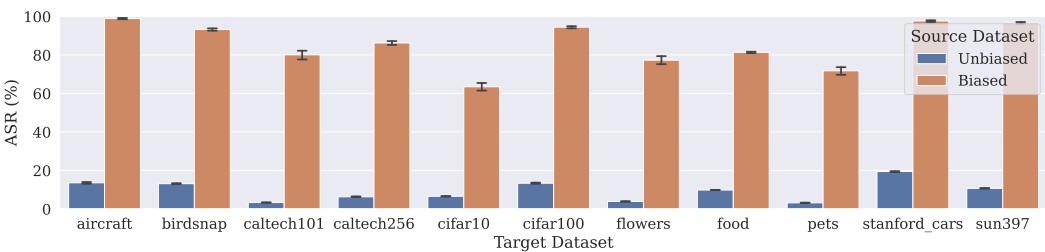

Figure 1: Bias consistently transfers across various target datasets in the fixed-feature transfer setting. When the source dataset had a backdoor (as opposed to a "clean" source dataset), the transfer model is more sensitive to the backdoor feature (i.e., ASR is higher). Error bars denote one standard deviation based on five random trials.

us to carefully vary (and study the effects of) properties of the bias such as how often it appears in the source dataset, how predictive it is of a particular label, and whether (and in what form) it also appears in the target dataset.

Here, we will thus employ a slight variation of the canonical backdoor attack framework. Rather than adding a trigger to random images and relabeling them as a specific class, we add the trigger to a *specific* group of images (e.g., 10% of the dogs in the source dataset) and leave the label unchanged. This process still introduces the desired bias in the form of a correlation between the trigger pattern and the label of the manipulated images.

**Experimental setup.** We focus our investigations on transfer learning from an (artificially modified) ImageNet-1K (Deng et al., 2009; Russakovsky et al., 2015) dataset to a variety of downstream target tasks[1]. Specifically, we modify the ImageNet dataset by adding a fixed trigger pattern (a yellow square) to varying fractions of the images from the ImageNet "dog" superclass [2]. Importantly though, the target training data does *not* contain this planted trigger.

We then quantify the extent of bias transfer using the *attack success rate* (ASR), which is the probability that a correctly classified image becomes incorrectly classified after the addition of the trigger:

$$\text{ASR}(\text{classifier } C, \text{trigger } T) = \Pr\left[C(T(x)) \neq y | C(x) = y\right], \tag{1}$$

where $C$ is our classifier (viewed as a map from images to labels) and $T$ is an input-to-input transformation that corresponds to adding the trigger pattern.

### 3.1 Bias consistently transfers in the fixed-feature transfer setting

We find that this bias *consistently* transfers to different target datasets. As in Gu et al. (2017), we begin with *fixed-feature* transfer learning, i.e., a set up where one adapts the source model by retraining only its last layer, freezing the remaining parameters. As Fig. 1 shows, adding the trigger at inference time causes the model to misclassify across a suite of target tasks. So clearly bias transfers in this setting. But how does the strength of the bias affect its transfer?

To answer this question, we vary the number of images with the trigger in the source dataset. Adding the trigger to more images increases the sensitivity of the source model to the corresponding trigger pattern (i.e., stronger bias)—see Fig. 2a. Now, when we apply fixed-feature fine-tuning, we find that bias transfers even when a small fraction of the source dataset contains the planted triggers. Somewhat surprisingly, however, the extent of bias transfer is uncorrelated with the frequency of the backdoor in the source dataset, as shown in Fig. 2b. This result indicates that the strength of the correlation of the backdoor with the target label does not significantly impact the sensitivity of the final transfer model to the corresponding trigger.

---

[1]We use the ResNet-18 architecture in the main paper, and study bias transfer on other architectures in Appendix A.4.

[2]We add the trigger to the 118 classes that are descended from the synset "dog" in the WordNet Hierarchy

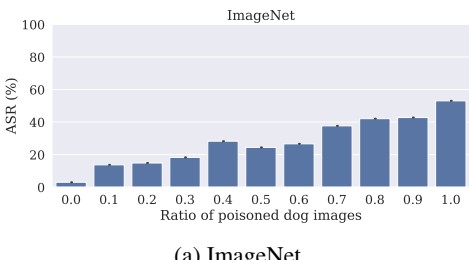
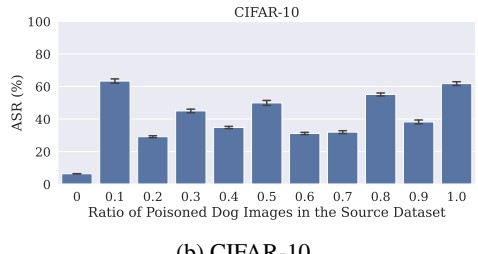

(a) ImageNet            (b) CIFAR-10

Figure 2: Attack Success Rate both on the source task with the original model (top) and on the target task with the transferred model (bottom). Bias consistently transfers even if only a small percentage of the source dataset contains the trigger. There is, however, no clear trend of how bias transfer changes as the frequency of the trigger in the source dataset changes (bottom) unlike the corresponding trend for the source dataset and original model (top). (Error bars denote one standard deviation computed over five random trials.)

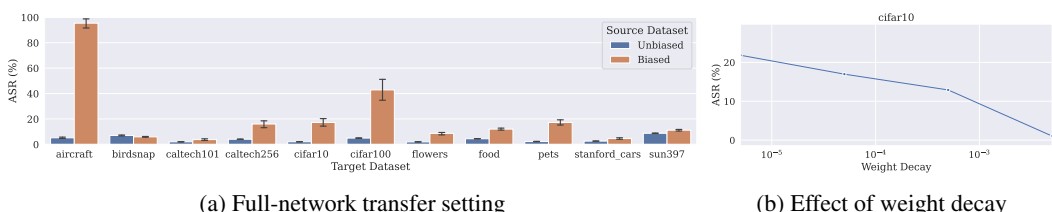

(a) Full-network transfer setting          (b) Effect of weight decay

Figure 3: Similarly to the fixed-feature setting, bias also transfers in the full-network setting, but to a lesser degree. (a) This holds consistently across various target datasets. Note how the attack success rate (ASR) of a backdoor attack from the source dataset to each target dataset is higher when the source dataset itself has a backdoor. (b) Observe also how increasing weight decay further reduces bias transfer (results for more datasets can be found in Appendix A.5). (Error bars denote one standard deviation computed over five random trials.)

## 3.2 FACTORS MITIGATING BIAS TRANSFER

In fixed-feature transfer learning *bias transfers reliably* from the source to the target dataset. Can we mitigate this bias transfer? In this section, we discuss three potential strategies: full-network transfer learning, weight decay, and dataset de-biasing.

**Can full-network transfer learning reduce bias transfer?** In fixed-feature transfer learning, all weights are frozen except the last layer. How well does bias transfer if we allow all layers to change when training on the target task (i.e., *full-network transfer learning*)? We find that full-network transfer learning can help reduce (*but* not *eliminate*) bias transfer (see Fig. 3a).

**Can weight decay mitigate bias transfer?** Weight decay is a natural candidate for reducing bias transfer; indeed, in our motivating logistic regression example from Section 2, weight decay eliminates the effect of any planted feature (see Appendix A.5 for a formal explanation). We find that increasing weight decay does *not* reduce bias transfer in the fixed-feature setting, but can *substantially* reduce bias transfer in the full-network transfer setting. Referring to Fig. 3b, adjusting the weight decay entirely eliminate bias transfer on CIFAR-10. However, the extent to which weight decay helps varies across datasets as we show in Appendix A.5.

**Can de-biasing (only) the target dataset remove the bias?** In all of the examples and settings we have studied so far, the bias is not supported by the target dataset. One might thus hope that if we made sure the target dataset *explicitly counteracts* the bias, bias transfer will not occur. This *de-biasing* can be expensive (and often unrealistic), as it requires prior knowledge of what biases need to be avoided, and then a way to embed these biases in the target dataset. But does it help?

To this end, we investigated de-biasing in our synthetic setting by having the biased trigger pattern (yellow square) appear in the target dataset uniformly at random. We found that, as shown in Fig. 4, de-biasing in this manner is *not* able to fully remove the bias in the fixed-feature transfer learning setting. However, in full-network transfer learning setting, the de-biasing intervention *does* succeed

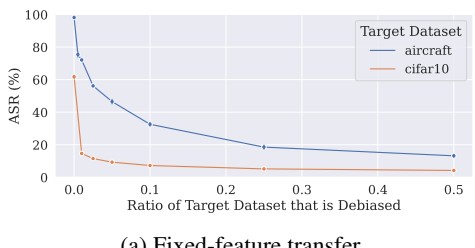 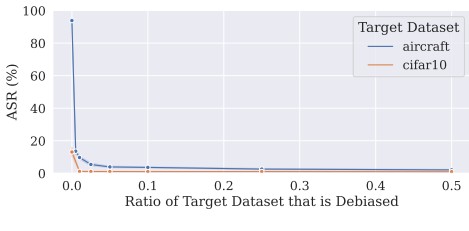

(a) Fixed-feature transfer          (b) Full-network transfer

Figure 4: (**left**) In the fixed-feature setting, de-biasing the target dataset by adding the trigger to uniformly across classes cannot fully prevent the bias from transferring. (**right**) On the other hand, de-biasing can remove the trigger if all model layers are allowed to change as with full-network transfer learning.

in correcting the bias. (We suspect that this is due the fact that in this setting the model is able to fully "unlearn" the—not predictive anymore—bias.)

Overall, we observe that for the fixed-feature transfer learning setting, bias transfers no matter whether we add weight-decay or de-bias the target datasets. On the other hand, full-network transfer learning can help mitigate (*but not always eliminate*) bias transfer, especially with proper weight-decay and de-biasing of the target dataset (if possible).

## 4 BIAS TRANSFER BEYOND BACKDOOR ATTACKS

In Section 3, we used synthetic backdoor triggers to show that biases can transfer from the source dataset (and, in the fixed-feature transfer setting, even when the target dataset is itself de-biased). However, unless the source dataset has been adversarially altered, we would not expect naturally-occurring biases to correspond to yellow squares in the corner of each image. Instead, these biases tend to be much more subtle, and revolve around issues such as over-reliance on image background (Xiao et al., 2020), or disparate accuracy across skin colors in facial recognition (Buolamwini & Gebru, 2018). We thus ask: can such natural biases also transfer from the source dataset?

As we demonstrate, this is indeed the case. Specifically, we study two such sample biases. First, we consider a *co-occurrence bias* between humans and dogs in the MS-COCO dataset (Lin et al., 2014). Then, we examine an *over-representation bias* in which models rely on gender to predict age in the CelebA dataset (Liu et al., 2015). In both cases, we modify the source task in order to amplify the effect of the bias, then observe that the bias remains even after fine-tuning on balanced versions of the dataset (in Section 5, we study bias transfer in a setting without such amplifications).

### 4.1 TRANSFERRING CO-OCCURRENCE BIASES IN OBJECT RECOGNITION

Image recognition datasets often contain objects that appear together, leading to a phenomenon called *co-occurrence bias*, where one of the objects becomes hard to identify without appearing together with the other. For example, since "skis" and "skateboards" typically occur together with people, models can struggle to correctly classify these objects without the presence of a person using them (Singh et al., 2020). Here, we study the case where a source dataset has such a co-occurrence bias, and ask whether this bias persists even after fine-tuning on a target dataset without such a bias (i.e., a dataset in which one of the co-occurring objects is totally absent).

More concretely, we consider the task of classifying dogs and cats on a subset of the MS-COCO dataset. We generate a *biased* source dataset by choosing images so that dogs (but not cats) always co-occur with humans (see Appendix A for the exact experimental setup), and we compare that with an unbiased source dataset that has no people at all. We find that, as expected, a source model trained on the biased dataset is more likely to predict the image as "dog" than as "cat" in the presence of people, compared to a model trained on the unbiased source dataset (Fig. 5a).[3]

---

[3]Note that the source model trained on the unbiased dataset seems to also be slightly sensitive to the presence of people even though it has never been exposed to any people. We suspect this is due to the presence of other confounding objects in the images.

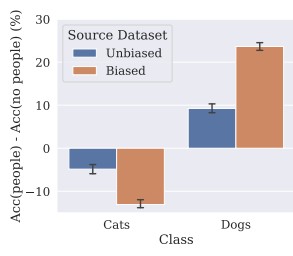 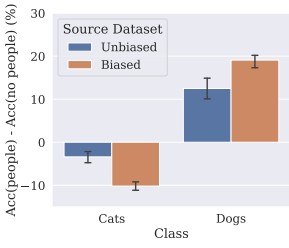 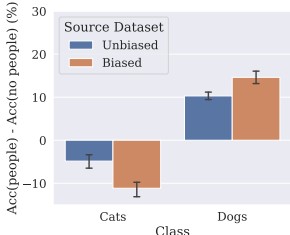

(a) Source Model          (b) Fixed-feature Transfer          (c) Full-network Transfer

Figure 5: **MS-COCO Experiment.** Bias transfer can occur when bias is a naturally occurring feature. We consider transfer from a source dataset that spuriously correlates the presence of dogs (but not cats) with the presence of people. We plot the difference in performance between images either contain or do not contain people. Even after fine-tuning on images without any people at all, models pre-trained on the biased dataset are highly sensitive to the presence of people.

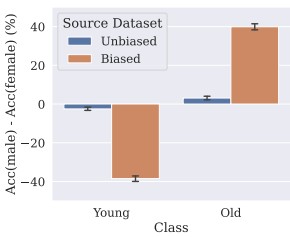 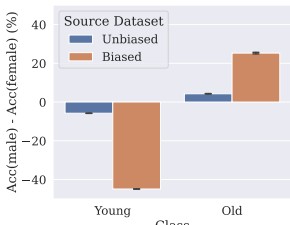 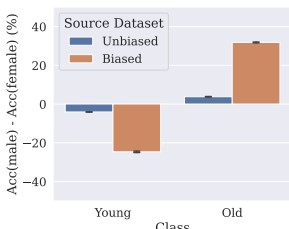

(a) Original source model    (b) Transfer on a target task    (c) Transfer on a target task:
                             containing only women           50% women and 50% men

Figure 6: **CelebA Experiment.** Bias transfer with natural features can occur even when the target dataset is de-biased. **(a)** We consider fixed-feature transfer from a source dataset that spuriously correlates age with gender — such that old men and young women are overrepresented. **(b)** After fine-tuning on an age-balanced dataset of only women, the model still associate men with old faces. **(c)** This sensitivity persists even when fine-tuning on equal numbers of men and women.

We then adapt this *biased* source model to a new target dataset that contains no humans at all, and check whether the final model is sensitive to the presence of humans. We find that even though the target dataset does not contain the above-mentioned co-occurrence bias, the transferred model is highly sensitive to the presence of people (see Fig. 5b). Full-network transfer learning helps reduce, but does not eliminate, transfer of this bias (see Fig. 5c).

### 4.2 Transferring gender bias in facial recognition

Facial recognition datasets are notorious for containing biases towards specific races, ages, and genders (Terhörst et al., 2021; Buolamwini & Gebru, 2018), making them a natural setting for studying bias transfer. For example, the CelebA dataset (Liu et al., 2015) over-represents subpopulations of older men and younger women. In this section, we use a CelebA subset that amplifies this bias, and pre-train source models on a source task of classifying "old" and "young" faces (we provide the exact experimental setup in Appendix A). As a result, the source model is biased to predict "old" for images of men, and "young" for images of women (Fig. 6a). Our goal is to study whether, after adapting this biased source model to a demographically balanced target dataset of faces, the resulting model will continue to use this spurious gender-age correlation.

To this end, we first adapt this biased source model on a dataset of exclusively female faces, with an equal number of young and old women. Here we consider fixed-feature transfer learning(and defer full-network transfer learning results to Appendix A). We then check if the resulting model still relies on "male-old" and "female-young" biases (Fig. 6b). It turns out that for both fixed-feature and full-network settings, these biases indeed persist: the downstream model is still more likely to predict "old" for an image of a male, and "young" for an image of a female.

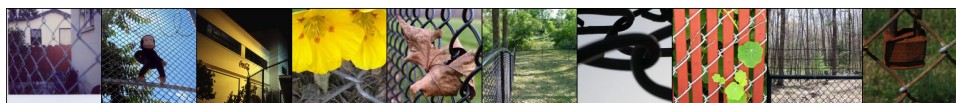

(a) Example images from the "Chain-link fence" class in ImageNet.

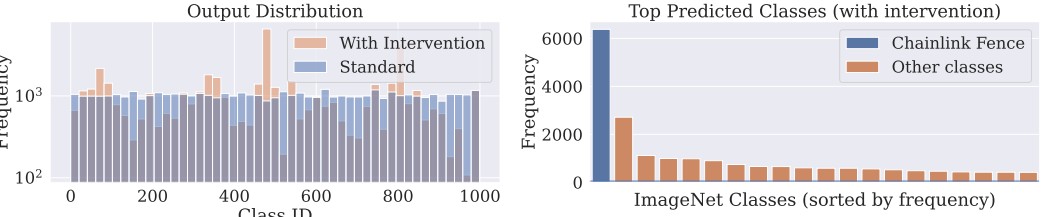

(b) Shift in ImageNet predicted class distribution after adding a chain-link fence intervention, establishing that the bias holds for the source model.

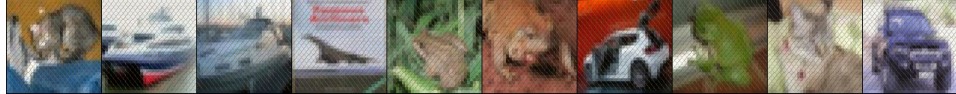

(c) Example CIFAR-10 images after applying the chain-link fence intervention.

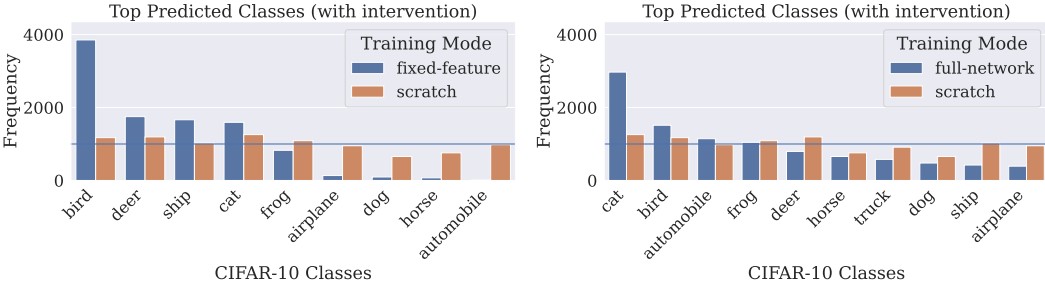

(d) Distribution of CIFAR-10 model predictions when trained from scratch and when transferred from the biased source model. We consider **(left)** fixed-feature and **(right)** full-network transfer learning. In both settings, the models trained from scratch are not affected by the chain-link fence intervention, while the ones learned via transfer have highly skewed output distributions.

Figure 7: **The "chainlink fence" bias. (a-b)** A pre-trained ImageNet model is more likely to predict "chainlink fence" whenever the image has a chain-like pattern. **(c-d)** This bias transfers to CIFAR-10 in both fixed-feature and full-network transfer settings. Indeed, if we overlay a chain-like pattern on all CIFAR-10 test set images, the model predictions skew towards a specific class. This does not happen if the CIFAR-10 model was trained from *scratch* instead (orange).

Can we remove this bias by adding images of men to the target dataset? To answer this question, we transfer the source model to a target dataset that contains equal numbers of men and women, balanced across both old and young classes (see Appendix A for other splits). We find that the transferred model is still biased (Fig. 6c), indicating that de-biasing the target task in this manner does not necessarily fix bias transfer.

## 5 BIAS TRANSFER IN THE WILD

In Section 4, we demonstrated that natural biases induced by subsampling standard datasets can transfer from source datasets to target tasks. We now ask the most advanced instantiation of our central question: do *natural* biases that *already exist* in the source dataset (i.e., where not enhanced by an intervention) also transfer?

To this end, we pinpoint examples of biases in the widely-used ImageNet dataset and demonstrate that these biases indeed transfer to downstream tasks (e.g., CIFAR-10), despite the latter not contain-

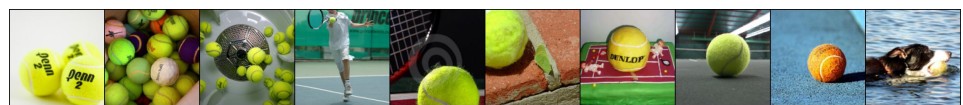

(a) Example images from the "tennis ball" class in ImageNet.

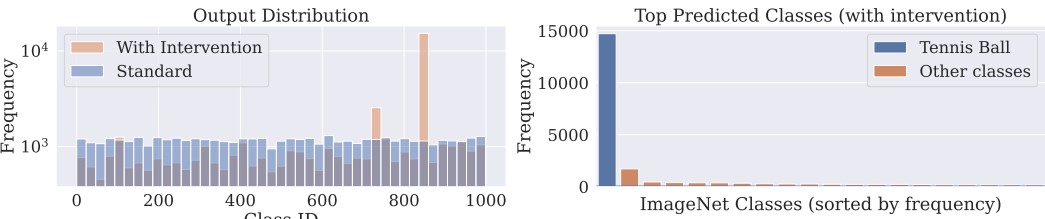

(b) Shift in ImageNet predicted class distribution after adding a tennis ball intervention, establishing that the bias holds for the source model.

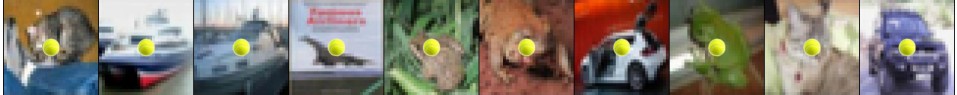

(c) Example CIFAR-10 images after applying the tennis ball intervention.

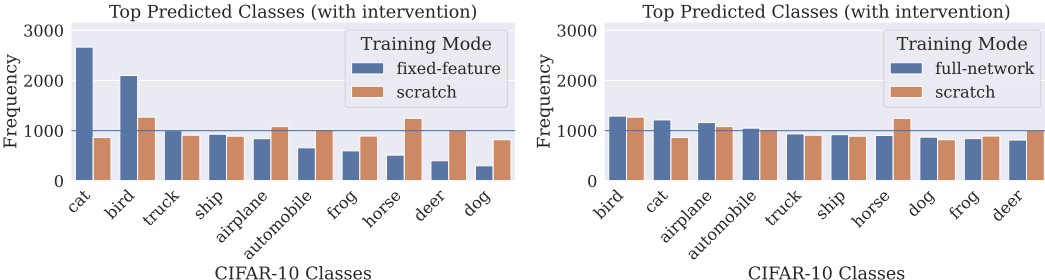

(d) Distribution of CIFAR-10 model predictions when trained from scratch and when transferred from the biased source model. We consider **(left)** fixed-feature and **(right)** full-network transfer learning. The from-scratch models are not affected by the tennis ball intervention, while the ones learned via transfer have highly skewed output distributions. Note that in this case, full-network transfer learning was able to remove the bias.

Figure 8: **The "tennis ball" bias. (a-b)** A pre-trained ImageNet model is more likely to predict "tennis ball" whenever a circular yellow shape is in the image. **(c-d)** This bias transfers to CIFAR-10 in the fixed-feature but not in the full network transfer settings.

ing such biases. Specifically, we examine here two such biases: the "chainlink fence" bias and the "tennis ball" bias (described below). Results for more biases and target datasets are in Appendix B.

**Identifying ImageNet biases.** To identify ImageNet biases, we focus on features that are (a) associated with an ImageNet class and (b) easy to overlay on an image. For example, we used a "circular yellow shape" feature is predictive for the class "tennis ball." To verify that these features indeed bias the ImageNet model, we consider a simple counterfactual experiment: we overlay the features on all the ImageNet images and monitor the shift in the model output distribution. As expected, both "circular yellow shape" and "chain-like pattern" are strong predictive features for the classes "tennis ball" and "chainlink fence"—see Fig. 7b and 8b. These naturally occurring ImageNet biases are thus suitable for studying the transfer of biases that exist in the wild.

**ImageNet-biases transfer to target tasks.** Now, what happens if we fine-tune a pre-trained ImageNet model (which has these biases) on a target dataset such as CIFAR-10? These biases turn out to persist in the resulting model even though CIFAR-10 does not contain them (as CIFAR-10 does not contain these classes). To demonstrate this phenomenon, we overlay the associated feature for both the "tennis ball" and "chainlink fence" ImageNet classes on the CIFAR-10 test set—see Fig. 7c

and 8c. We then evaluate (1) a model fine-tuned on a standard pre-trained ImageNet model, and (2) a model trained from scratch on the CIFAR-10 dataset.

As Fig. 7d-(left) and 8d-(left) demonstrate, the fine-tuned models using fixed-feature transfer learning are sensitive to the overlaid ImageNet biases, whereas CIFAR-10 models trained from scratch are not. This is corroborated by the overall skew of the output class distribution for the transfer-learned model, compared to an almost uniform output class distribution of the model trained from scratch. Note that, as mentioned in Section 3.2, full-network transfer learning can sometimes mitigate bias transfer, which we observe for the "tennis ball" bias in Fig. 8d-(right). Though for other biases, as shown in Fig. 7d-(right) and Appendix B, the bias effect persists even after full-network fine-tuning.

## 6 RELATED WORK

**Transfer learning.** Transfer learning has been used in applications ranging from autonomous driving (Kim & Park, 2017; Du et al., 2019), radiology (Wang et al., 2017; Ke et al., 2021) to satellite image analysis (Xie et al., 2016; Wang et al., 2019). In particular, fine-tuning pre-trained ImageNet models has increasingly become standard practice to improve the performance of various image classification (Kornblith et al., 2019; Salman et al., 2020; Utrera et al., 2020) and object detection tasks (Ren et al., 2015; Dai et al., 2016; Girshick et al., 2014; Chen et al., 2017). More recently, even larger vision (Radford et al., 2021; Sun et al., 2017) and language models (Brown et al., 2020)—often trained on proprietary datasets—have acted as backbones for downstream tasks. With this widespread usage of pre-trained models, it is important to understand whether any limitation of these models would affect downstream tasks, which is what we focus on in this work.

There has been some work to understand the types of features that are transferred in transfer learning (Neyshabur et al., 2020; Jain et al., 2022). Shafahi et al. (2019) further finds that adversarial robustness can also transfer during fine-tuning.

**Backdoor attacks.** In a backdoor attack (Gu et al., 2017; Evtimov et al., 2018; Turner et al., 2019), an adversary maliciously injects a trigger into the source dataset which can be activated during inference. This type of attack can be especially hard to detect, since the model performs well in the absence of the trigger. Indeed, there exists a long line of work on injecting malicious training examples, known as data poisoning attacks (Biggio et al., 2012; Xiao et al., 2012; Newell et al., 2014; Mei & Zhu, 2015; Steinhardt et al., 2017). Gu et al. (2017) planted backdoors in a stop-sign detection dataset, and found that fine-tuned stop-sign detection models were still sensitive to this trigger. Related to our work, Lubana et al. (2023) perform a mechanistic analysis where they fine-tune from a backdoored CIFAR-10 to a clean CIFAR-10 and find that the model is still biased.

**Catastrophic forgetting.** Bias transfer can be seen as the inverse of catastrophic forgetting (Kirkpatrick et al., 2017), a well known phenomena (Hinton & Plaut, 1987) in which models trained on different tasks sequentially can "forget" how to solve earlier tasks, resulting in lower accuracies. Model behavior around bias transfer is consistent with catastrophic forgetting: the degree of catastrophic forgetting (or bias transfer) depends on both the training algorithm used and on the task setup. For example, just as how training on sequential tasks can yield poor model accuracy on earlier tasks, full-network fine-tuning (especially with high weight decay, cf. Figure 3b) can reduce biases imbued by network pretraining. One fruitful area of future work could apply principles from catastrophic forgetting and continual learning (De Lange et al., 2022) to mitigate bias transfer.

## 7 CONCLUSION

In this work we demonstrated that biases in pre-trained models tend to remain present even after fine-tuning these models on downstream target tasks. Crucially, these biases can persist even when the target dataset used for fine-tuning did not contain such biases.

While we suggest several mitigation strategies—such as combining full-network fine-tuning with high weight decay or de-biasing—these approaches often do not fully prevent bias transfer. These findings are of particular concern as researchers and practitioners increasingly leverage public pre-trained source models, which are likely to contain undocumented biases. We thus encourage further investigation of the full ML pipeline—even parts that are seemingly unimportant—for potential sources of bias.

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

# A  EXPERIMENTAL SETUP

## A.1  IMAGENET MODELS

In this paper, we train a number of ImageNet models and transfer them to various datasets in Sections 3 and 5. We mainly use the ResNet-18 architecture all over the paper. However, we study bias transfers using various architectures in Appendix A.4. We use PyTorch's official implementation for these architectures, which can be found here https://pytorch.org/vision/stable/models.html.

**Training details.**  We train our ImageNet models from scratch using SGD by minimizing the standard cross-entropy loss. We train for 16 epochs using a Cyclic learning rate schedule with an initial learning rate of $0.5$ and learning rate peak epoch of $2$. We use momentum of $0.9$, batch size of $1024$, and weight decay of $5e^{-4}$. We use standard data-augmentation: *RandomResizedCrop* and *RandomHorizontalFlip* during training, and *RandomResizedCrop* during testing. Our implementation and configuration files are available in the attached code.

## A.2  TRANSFER DETAILS FROM IMAGENET TO DOWNSTREAM IMAGE CLASSIFICATION TASKS

**Transfer datasets.**  We use the image classification tasks that are used in (Salman et al., 2020; Kornblith et al., 2019), which have various sizes and number of classes. When evaluating the performance of models on each of these datasets, we report the Top-1 accuracy for balanced datasets and the Mean Per-Class accuracy for the unbalanced datasets. See Table 1 for the details of these datasets. For each dataset, we consider two transfer learning settings: *fixed-feature* and *full-network* transfer learning which we describe below.

Table 1: Image classification benchmarks used in this paper. Accuracy metric is the metric we report for each of the dataset across the paper. Some datasets are imbalanced, so we report Mean Per-Class accuracy for those. For the rest, we report Top-1 accuracy.

| Dataset | Size (Train/Test) | Classes | Accuracy Metric |
|---|---|---|---|
| Birdsnap (Berg et al., 2014) | 32,677/8,171 | 500 | Top-1 |
| Caltech-101 (Fei-Fei et al., 2004) | 3,030/5,647 | 101 | Mean Per-Class |
| Caltech-256 (Griffin et al., 2007) | 15,420/15,187 | 257 | Mean Per-Class |
| CIFAR-10 (Krizhevsky, 2009) | 50,000/10,000 | 10 | Top-1 |
| CIFAR-100 (Krizhevsky, 2009) | 50,000/10,000 | 100 | Top-1 |
| FGVC Aircraft (Maji et al., 2013) | 6,667/3,333 | 100 | Mean Per-Class |
| Food-101 (Bossard et al., 2014) | 75,750/25,250 | 101 | Top-1 |
| Oxford 102 Flowers (Nilsback & Zisserman, 2008) | 2,040/6,149 | 102 | Mean Per-Class |
| Oxford-IIIT Pets Parkhi et al. (2012) | 3,680/3,669 | 37 | Mean Per-Class |
| SUN397 (Xiao et al., 2010) | 19,850/19,850 | 397 | Top-1 |
| Stanford Cars (Krause et al., 2013) | 8,144/8,041 | 196 | Top-1 |

**Fixed-feature transfer.**  For this setting, we *freeze* the layers of the ImageNet source model[4], except for the last layer, which we replace with a random initialized linear layer whose output matches the number of classes in the transfer dataset. We now train only this new layer for using SGD, with a batch size of 1024 using cyclic learning rate. For more details and hyperparameter for each dataset, please see config files in the attached code.

**Full-network transfer.**  For this setting, we *do not freeze* any of the layers of the ImageNet source model, and all the model weights are updated. We follow the exact same hyperparameters as the fixed-feature setting.

---

[4]We do not freeze the batch norm statistics, but only the weights of the model similar to (Salman et al., 2020).

**Compute and training time**   Throughout the paper, we use the FFCV data-loading library to train models fast  (Leclerc et al., 2022). Using FFCV, we can train an ImageNet model, for example, in around 1 hr only on a single V100 GPU. Our experiments were conducted on a GPU cluster containing A100 and V100 GPUs.

## A.3   DEFINITIONS

We define biases formally as follows. For input $x \in \mathcal{X}$, let $Y$ be a random variable corresponding to the true label, and let $\hat{Y}$ be a random variable corresponding to the predicted label.

We can define a "feature" as a transformation $T : \mathcal{X} \to \mathcal{X}$ which applies some property to an input $x$. For example, if the feature is "black background", the transformation $T$ would apply a black background to $X$. We say that example $x$ "has" a feature $T$ if $x = T(x)$ (adding the feature did not change the input).

We say that feature $T$ is a $\delta$ "bias" for a model if:

$$\mathrm{P}(\hat{Y}|T(x)) - \mathrm{P}(\hat{Y}|x) > \delta + \mathrm{P}(Y|T(x)) - \mathrm{P}(Y|x).$$

That is, if the predicted label is more sensitive to the addition of the feature than the true label.

The phenomenon of bias transfer is when $T$ is a $\delta$-bias for a model pre-trained on a source dataset, but is not a $\delta$-bias for a model trained from scratch (i.e., without pre-training).

## A.4   VARYING ARCHITECTURES

In this section, we study whether bias transfers when applying transfer learning using various architectures. We conduct the basic experiment of Section 3 on several standard architectures from the PyTorch's Torchvision[5].

As in Section 3, we train two versions of each architecture: one on a clean ImageNet dataset, and another on a modified ImageNet dataset containing a backdoor. We use the same hyperparameters as the rest of the paper, except for the batch size, which we set to 512 instead of 1024. The reason we lower the batch size is to fit these models in memory on a single A100 GPU.

Now, we transfer each of these models to a clean CIFAR-10 dataset, and test if the backdoor attack transfers. Similar to the results of the main paper, we notice that backdoor attack indeed transfers in the fixed-feature setting. We note however that for the full-network setting, all architectures other than ResNet-18 (which we use in the rest of the paper) seem to be more robust to the backdoor attack.

We further repeat the backdoor experiments for three sizes of Vision Transformer (Figure 10). We find that bias transfer still occurs in both the fixed-feature and full-network settings.

## A.5   THE EFFECT OF WEIGHT DECAY IN FULL-NETWORK TRANSFER LEARNING

As mentioned in Section 3.2, we found weight decay to have a significant impact on bias transfer in the full-network transfer learning setting. In particular, increasing weight decay reduces bias transfer. Here, we present a formal explanation of why this happens by studying this within the logistic regression example we presented in Section 2. Recall that, following the setup in Section 2, if we transfer a pre-trained linear classifier $\boldsymbol{w}_{src}$ to a target dataset $\{(\boldsymbol{x}_i, y_i)\}$, $\boldsymbol{w}_{src}$ is preserved in all directions orthogonal to the span of the $\boldsymbol{x}_i$.

Now what happens if we add $\ell_2$ regularization (i.e., weight decay) to the logistic regression problem? As can be easily checked, the gradient updated of the logistic loss now becomes

$$\nabla \ell_{\boldsymbol{w}}(\boldsymbol{x}_i, y_i) = (\sigma(\boldsymbol{w}^\top \boldsymbol{x}_i) - y_i) \cdot \boldsymbol{x}_i + \lambda \boldsymbol{w}, \tag{2}$$
$$= (\sigma(\boldsymbol{w}_S^\top \boldsymbol{x}_i) - y_i) \cdot \boldsymbol{x}_i + \lambda(\boldsymbol{w}_S + \boldsymbol{w}_{S'})$$

where $\lambda$ is the regularization strength, $\boldsymbol{w}_S$ and $\boldsymbol{w}_{S'}$ are the projections of $\boldsymbol{w}$ on the span of the target data points $\boldsymbol{x}_i$'s, denoted $S$, and on its complementary subspace, denoted $S'$. This gradient,

---

[5]These models can be found here https://pytorch.org/vision/stable/models.html

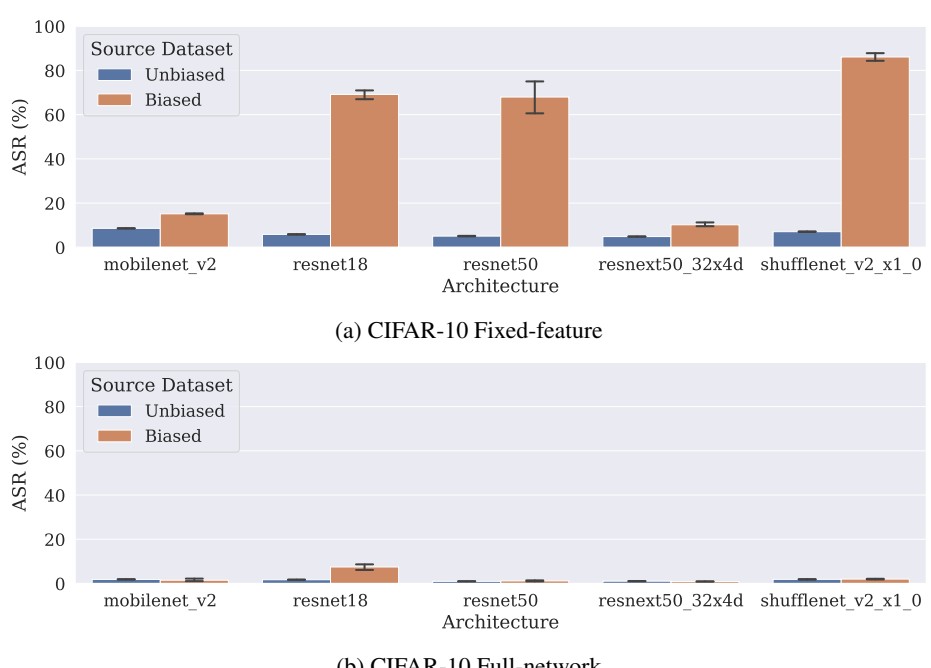

(a) CIFAR-10 Fixed-feature

(b) CIFAR-10 Full-network

Figure 9: Backdoor attack (bias) consistently transfers in the fixed-feature setting across various architectures. However, this happens to a lesser degree in the full-network transfer setting.

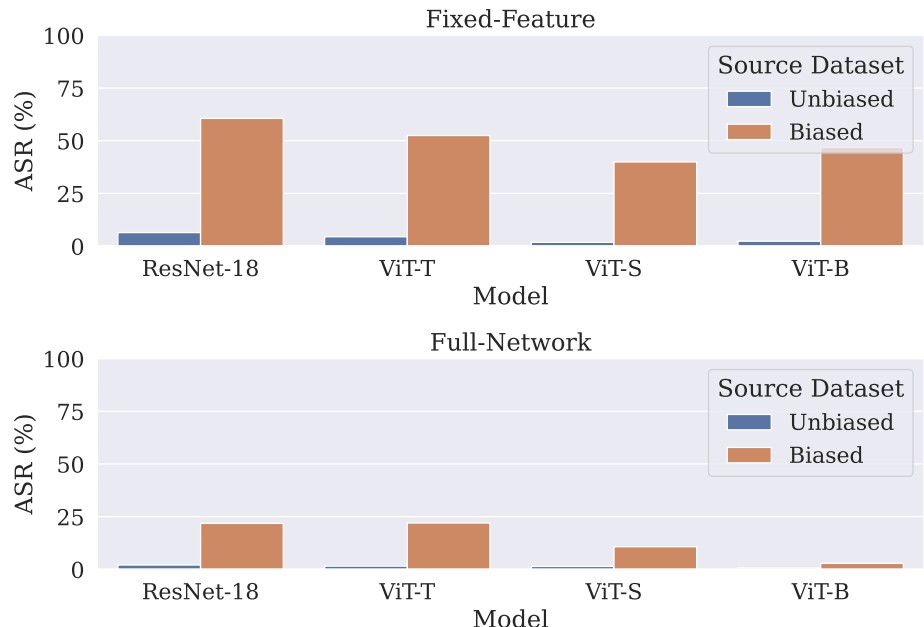

Figure 10: We repeat the backdoor experiments for three sizes of ViT. We find that bias transfer still occurs in both the fixed-feature and full-network settings.

as before, restricts the space of updates to those in $S$. However due to regularization, this gradient drives $\boldsymbol{w}_{S'}$ to zero. Therefore, any planted bias in $S'$ disappears as this subspace collapses to zero with regularization.

We illustrate this example in a toy setting, where the dataset contains 1000 points (dimension 4). The first two dimensions are drawn from two normal distributions ($\mathcal{N}(\pm 2, 1)$) depending on the

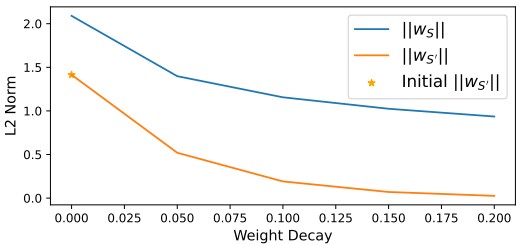

Figure 11: $||\mathbf{w}_S||_2$ and $||\mathbf{w}_{S'}||_2$ for varying weight decays. While $||\mathbf{w}_{S'}||_2$ is unchanged from the original weight vector when weight decay is 0, it is gradually driven to 0 as weight decay increases.

class. However the last two dimensions are set to 0 in the training dataset. We then train a logistic regression classifier with SGD with varying weight decay, where the initial weight vector is set to all 1s. Here $\mathbf{w}_S$ are the first two dimensions of the weight vector, and $\mathbf{w}_{S'}$ are the second two dimensions. We plot $||\mathbf{w}_S||_2$ and $||\mathbf{w}_{S'}||_2$ for varying weight decays (see Figure 11.) While $||\mathbf{w}_{S'}||_2$ is unchanged from the original weight vector when weight decay is 0, it is gradually driven to 0 as weight decay increases.

Indeed, we observe in practice that as we increase weight decay in the full-network transfer learning regime, bias transfer decreases over various downstream tasks as shown in Figure 12. On the other hand, we find that weight decay does not reduce bias transfer in the fixed feature transfer learning regime, where the weights of the pretrained model are frozen.

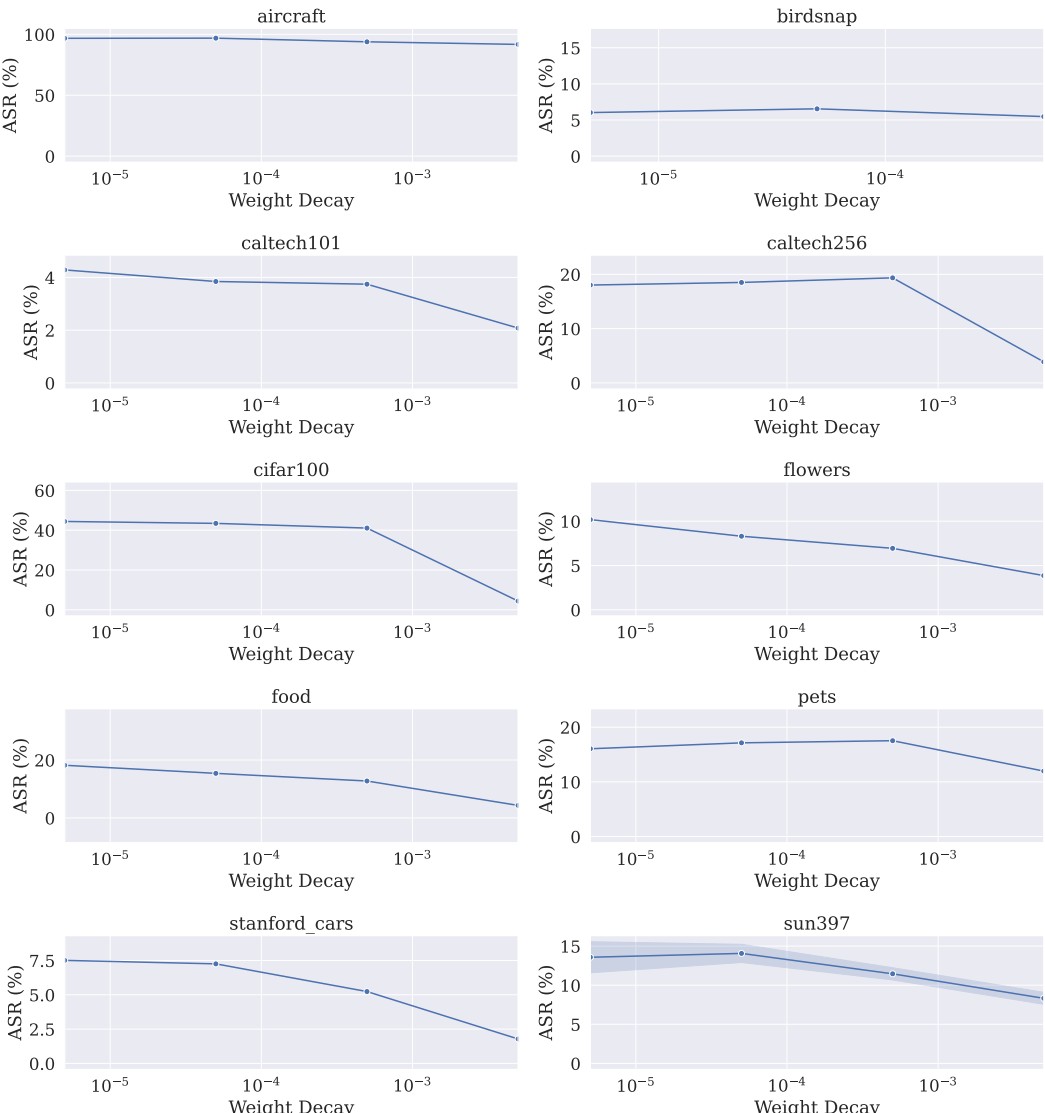

Figure 12: As weight decay increases, the ASR decreases which means bias transfers less across various datasets. We increase weight decay until the clean accuracy on the target dataset significantly deteriorates (see Figure 13). Error regions correspond to standard deviation over five random trials.

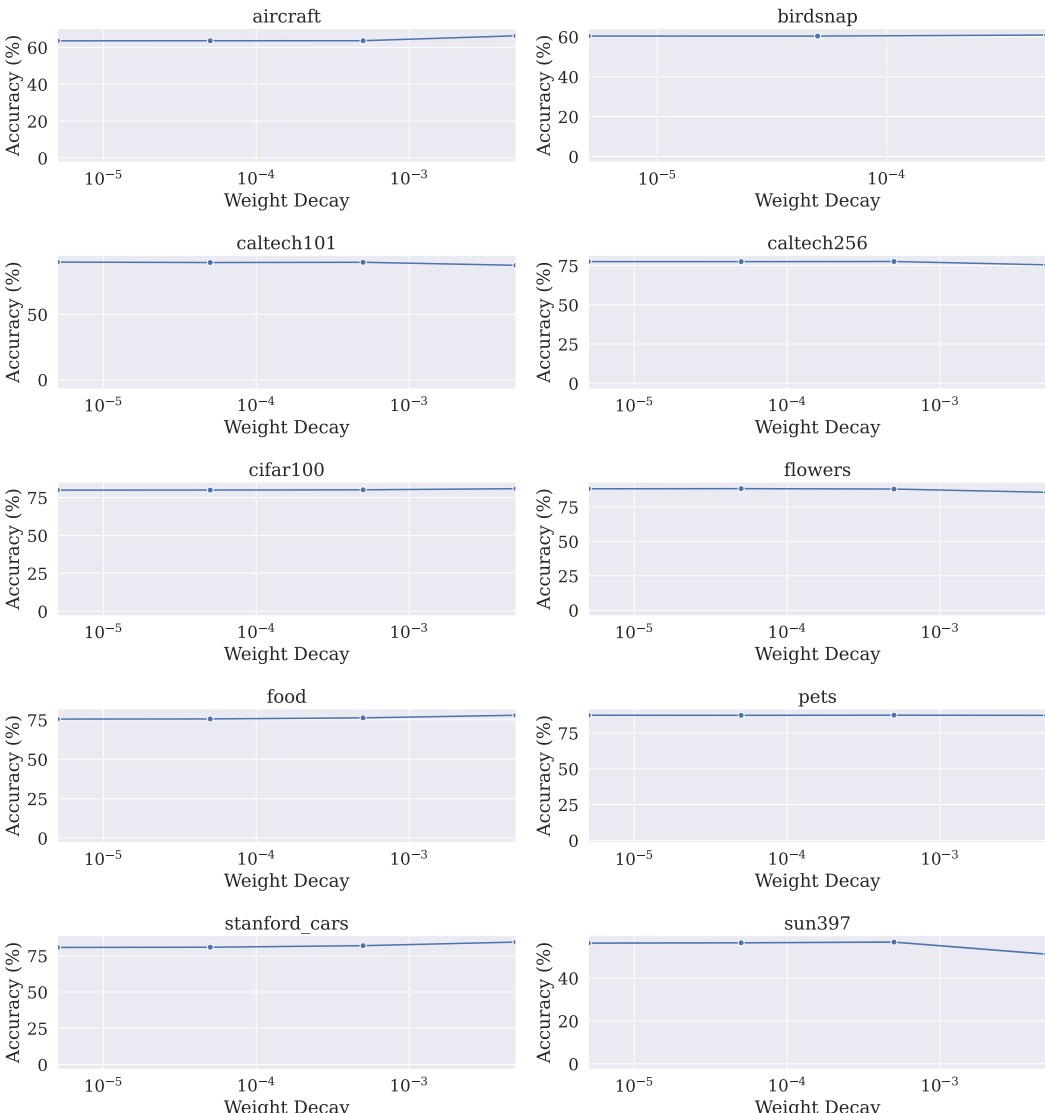

Figure 13: The clean accuracies corresponding to the weight-decay experiment. We increase weight decay as long as the clean accuracy on the target dataset is roughly the same. Error regions (very small) correspond to standard deviation over five random trials.

### A.6 CLEAN ACCURACIES FOR EXPERIMENTS OF SECTION 3

In Figure 14, we report the clean accuracies of the transferred models that we use Section 3 on various target datasets. Note how the accuracies of both models pretrained in biased and unbiased source models, for both fixed-feature and full-network settings, are roughly the same. So the discrepancy in ASR reported in the main paper is solely due to bias transfer.

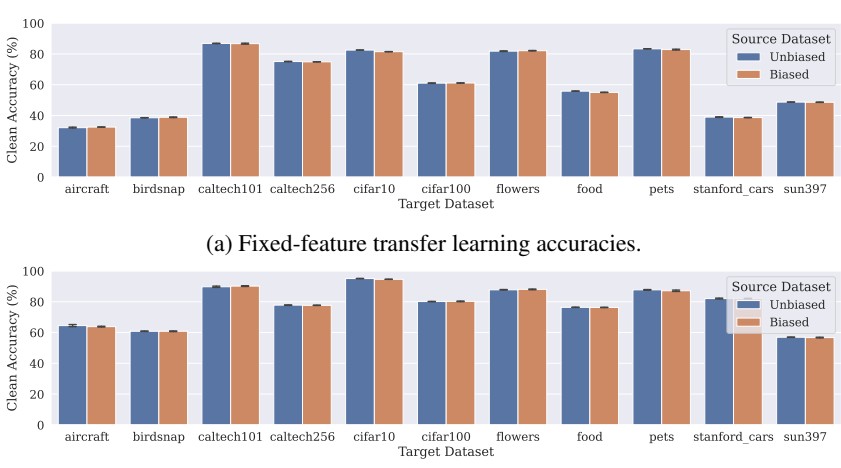

(a) Fixed-feature transfer learning accuracies.

(b) Full-network transfer learning accuracies.

Figure 14: Clean accuracies of the fixed-feature and full-network experiments of Section 3.

### A.7 COMPARISON WITH MODELS TRAINED FROM SCRATCH (ADDITIONAL RESULTS TO SECTION 3)

In this section, we add an extra baseline to Figure 3a where we train models from scratch on the various target datasets to check if the yellow square bias already exists in these datasets. In Figure 15a, we plot the accuracies of all the models across all target tasks. Note that since there is a significant difference between the accuracies of the models trained from scratch and those finetuned, ASR is no longer an informative metric to capture the existence of bias. Thus, we measure the change in accuracy after adding the backdoor trigger and report the results in Figure 15b. Indeed, the addition of the yellow square trigger do not significantly change the accuracy of the models trained from scratch reflecting no existing bias in the target datasets.

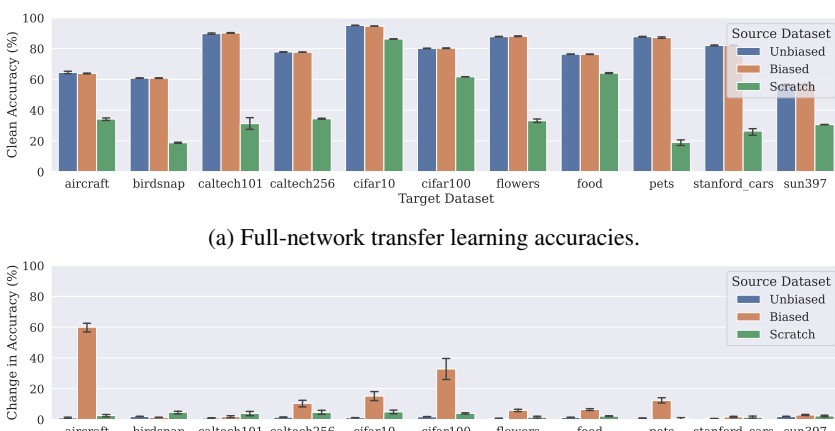

(a) Full-network transfer learning accuracies.

(b) Full-network transfer learning change in accuracy after adding the backdoor trigger.

Figure 15: Additional baseline ("Scratch") for the experiment of Figure 3a.

### A.8   THE EFFECT OF LEARNING RATE ON BIAS TRANSFER

We further analyze the impact of the fine-tuning learning rate on bias transfer. In Figure 16 repeat the ImageNet to CIFAR-10 experiment with a yellow square, but now increase and decrease the learning rate (from their default of 0.1). Like increasing weight decay, increasing the learning rate actually slightly worsens bias transfer for the fixed-feature setting, while helping mitigate in the full-network setting. However, changing the learning rate is a relatively invasive intervention, which may have impacts on the model's overall accuracy.

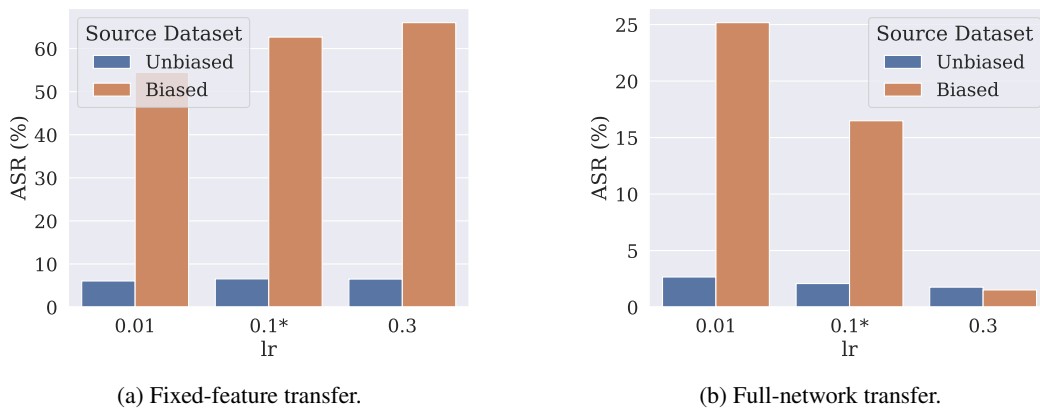

(a) Fixed-feature transfer.                    (b) Full-network transfer.

Figure 16: ASR when varying the fine-tuning learning rate for fixed-feature and full-network transfer.

### A.9   BIAS TRANSFER CAN AMPLIFY EXISTING BIASES.

In this section, we consider the case where the downstream dataset is also biased. Specifically, we add a yellow square to the dog class of the downstream CIFAR-10 training dataset (at varying frequencies). Indeed, we find the pre-training on a biased dataset amplifies the downstream models' sensitivity to the yellow square (see Figure 17).

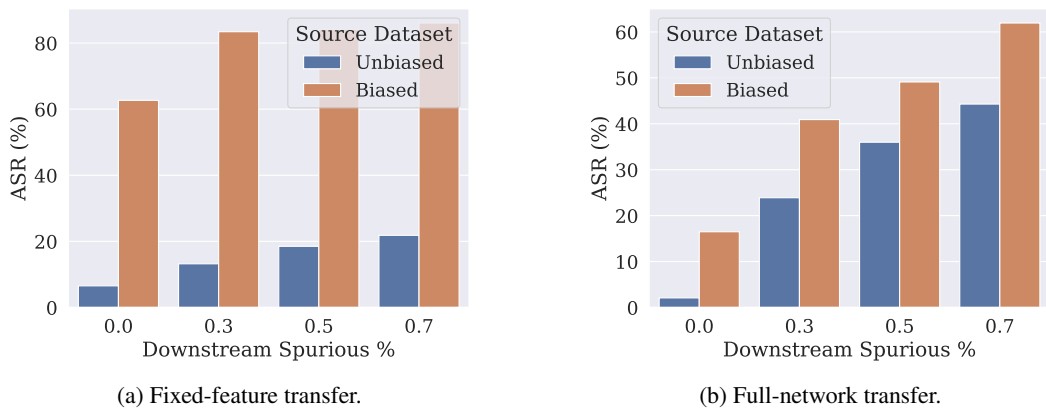

(a) Fixed-feature transfer.                    (b) Full-network transfer.

Figure 17: ASR when the downstream task is also biased.

## A.10 MS-COCO

In this section, we provide experimental details for the experiment on MS-COCO in Section 4.1. We consider the binary task of predicting cats from dogs, where there is a strong correlation between dogs and the presence of people.

**Dataset construction.** We create two source datasets which are described in Table 2.

Table 2: The synthetic datasets we create from MS-COCO for the experiment in Section 4.1.

| Dataset | Class: Cat | | Class: Dog | |
|---|---|---|---|---|
| | With People | Without People | With People | Without People |
| Non-Spurious | 0 | 1000 | 0 | 100 |
| Spurious | 1000 | 4000 | 4000 | 1000 |

We then fine-tune models trained on the above source datasets on new images of cats and dogs without people (485 each). We use the cats and dogs from the MS-COCO test set for evaluation.

**Experimental details.** We train a ResNet-18 with resolution $224 \times 224$. We use SGD with momentum, and a Cyclic learning rate. We use the following hyperparameters shown in Table 3:

Table 3: Hyperparameters used for training on the MS-COCO dataset.

| Hyperparameter | Value for pre-training | Value for fine-tuning |
|---|---|---|
| Batch Size | 256 | 256 |
| Epochs | 25 | 25 |
| LR | 0.01 | 0.005 |
| Momentum | 0.9 | 0.9 |
| Weight Decay | 0.00005 | 0.00005 |
| Peak Epoch | 2 | 2 |

## A.11 CELEBA

In this section, we provide experimental details for the CelebA experiments in Section 4.2. Here, the task was to distinguish old from young faces, in the presence of a spurious correlation with gender in the source dataset.

**Dataset construction.** We create two source datasets shown in Table 4:

Table 4: The synthetic source datasets we create from CelebA for the experiment in Section 4.2.

| Dataset | Class: Young | | Class: Old | |
|---|---|---|---|---|
| | Male | Female | Male | Female |
| Non-Spurious | 2500 | 2500 | 2500 | 2500 |
| Spurious | 1000 | 4000 | 4000 | 1000 |

Due to imbalances in the spurious dataset, the model trained on this dataset struggles on faces of young males and old females. We then fine-tune the source models on the following target datasets (see Table 5), the images of which are disjoint from that in the source dataset.

Due to space constraints, we plotted the results of fixed-feature transfer setting on Only Women and 80% Women|20% Men in the main paper. Below, we display the results for fixed-feature and full-network transfer settings on all 3 target datasets.

**Experimental details.** We train a ResNet-18 with resolution $224 \times 224$. We use SGD with momentum, and a cyclic learning rate. We use the following hyperparameters shown in Table 6:

Table 5: The synthetic target datasets we create from CelebA for the experiment in Section 4.2.

| Dataset | Class: Young | | Class: Old | |
| --- | --- | --- | --- | --- |
| | Male | Female | Male | Female |
| Only Women | 0 | 5000 | 0 | 5000 |
| 80% Women\|20% Men | 1000 | 4000 | 1000 | 4000 |
| 50% Women\|50% Men | 2500 | 2500 | 2500 | 2500 |

Table 6: Hyperparameters used for training on the CelebA datasets.

| Batch Size | Epochs | LR | Momentum | Weight Decay | Peak Epoch |
| --- | --- | --- | --- | --- | --- |
| 1024 | 20 | 0.05 | 0.9 | 0.01 | 5 |

**Results.** We find that in both the fixed-feature and full-network transfer settings, the gender correlation transfers from the source model to the transfer model, even though the target task is itself gender balanced as shown in Figure 18. As the proportion of men and women in the target dataset change, the model is either more sensitive to the presence of women, or more sensitive to the presence of men. In all cases, however, the model transferred from the spurious backbone is more sensitive to gender than a model transferred from the non-spurious backbone.

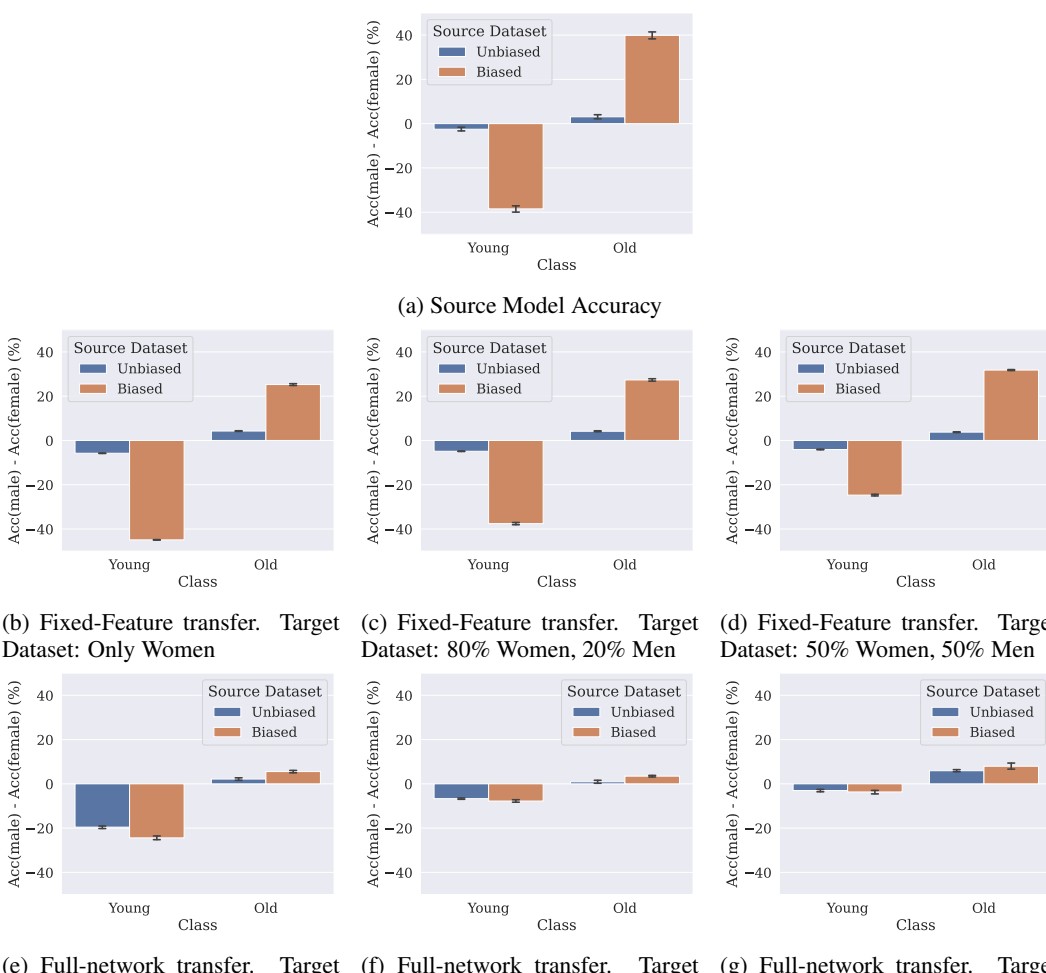

(a) Source Model Accuracy

(b) Fixed-Feature transfer. Target Dataset: Only Women

(c) Fixed-Feature transfer. Target Dataset: 80% Women, 20% Men

(d) Fixed-Feature transfer. Target Dataset: 50% Women, 50% Men

(e) Full-network transfer. Target Dataset: Only Women

(f) Full-network transfer. Target Dataset: 80% Women, 20% Men

(g) Full-network transfer. Target Dataset: 50% Women, 50% Men

Figure 18: **CelebA Experiment.** We consider transfer from a source dataset that spuriously correlate age with gender — such that old men and young women are overrepresented. We plot the difference in accuracies between male and female examples, and find that the model transferred from a spurious backbone is sensitive to gender, even though the target dataset was itself gender balanced.

# B  IMAGENET BIASES

## B.1  CHAINLINK FENCE BIAS.

In this section we show the results for the "chainlink fence" bias transfer. We first demonstrate in Figure 19 that the "chainlink fence" bias actually exists in ImageNet. Then in Figures 20, 21, 22, and 23, we show the output distribution—after applying a chainlink fence intervention—of models trained on various datasets either from scratch, or by transferring from the ImageNet model. The from-scratch models are not affected by the chainlink fence intervention, while the ones learned via transfer have highly skewed output distributions.

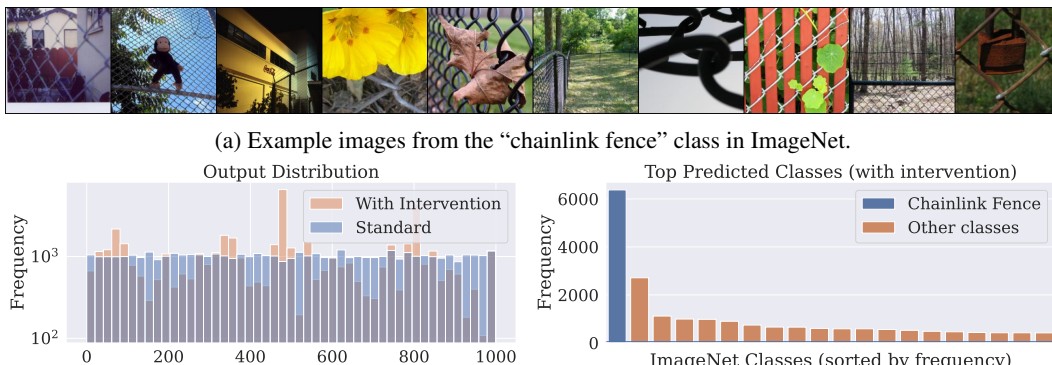

(a) Example images from the "chainlink fence" class in ImageNet.

(b) Shift in ImageNet predicted class distribution after adding a "chainlink fence" intervention, establishing that the bias holds for the source model.

Figure 19: The **chainlink fence** bias in ImageNet.

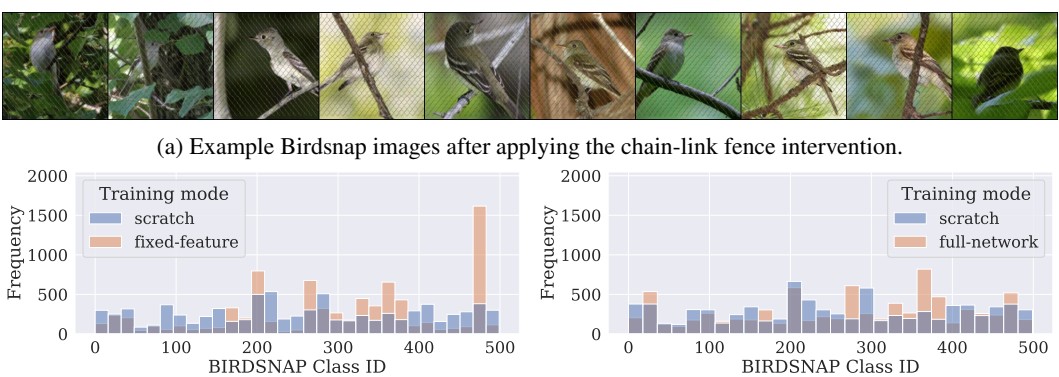

(a) Example Birdsnap images after applying the chain-link fence intervention.

(b) Output distribution of Birdsnap models with a chainlink fence intervention.

Figure 20: The **chainlink fence** bias transfers to *Birdsnap*.

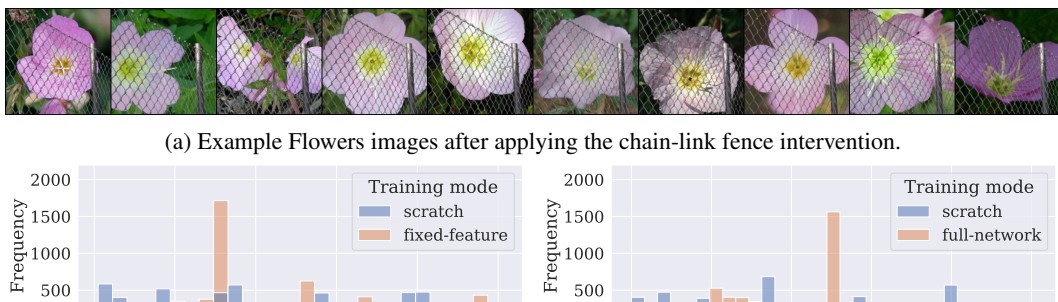

(a) Example Flowers images after applying the chain-link fence intervention.

(b) Output distribution of Flowers models with a chainlink fence intervention.

Figure 21: The **chainlink fence** bias transfers to *Flowers*.

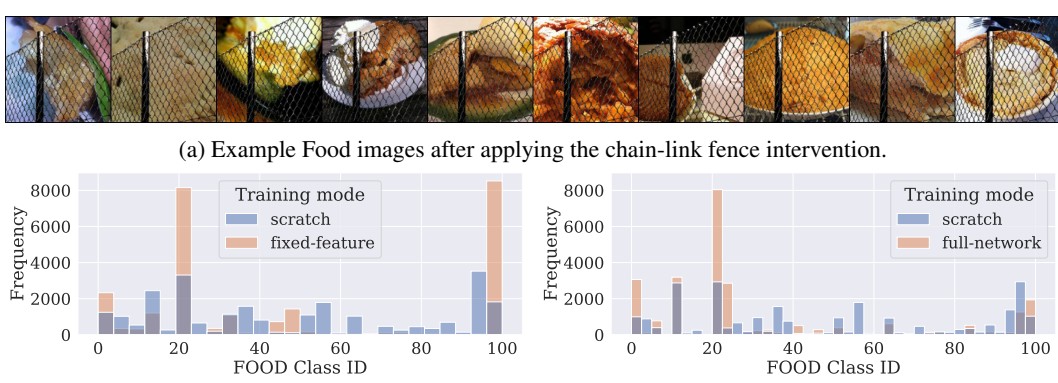

(a) Example Food images after applying the chain-link fence intervention.

(b) Output distribution of Food models with a chainlink fence intervention.

Figure 22: The **chainlink fence** bias transfers to *Food*.

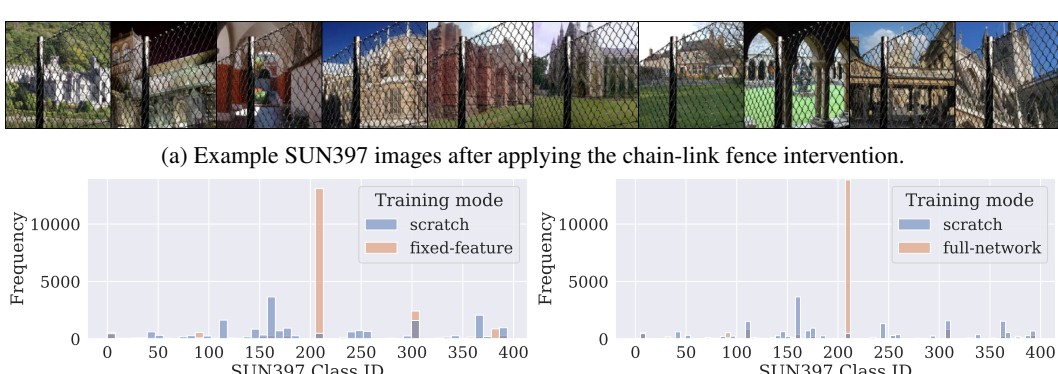

(a) Example SUN397 images after applying the chain-link fence intervention.

(b) Output distribution of SUN397 models with a chainlink fence intervention.

Figure 23: The **chainlink fence** bias transfers to *SUN397*.

## B.2 HAT BIAS.

In this section we show the results for the "Hat" bias transfer. We first demonstrate in Figure 24 that the "Hat" bias actually exists in ImageNet (shifts predictions to the "Cowboy hat" class). Then in Figure 25, we show the output distribution—after applying a hat intervention—of models trained on CIFAR-10 either from scratch, or by transferring from the ImageNet model. The from-scratch model is not affected by the hat intervention, while the one learned via transfer have highly skewed output distributions.

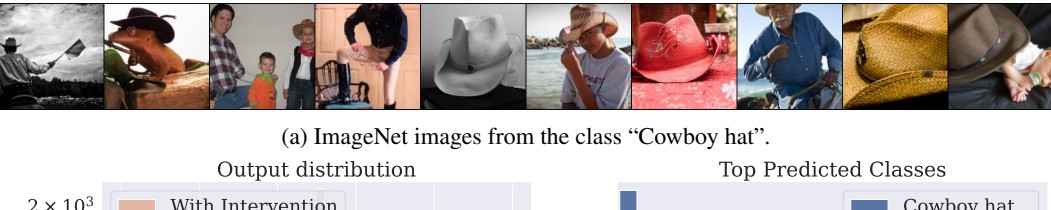

(a) ImageNet images from the class "Cowboy hat".

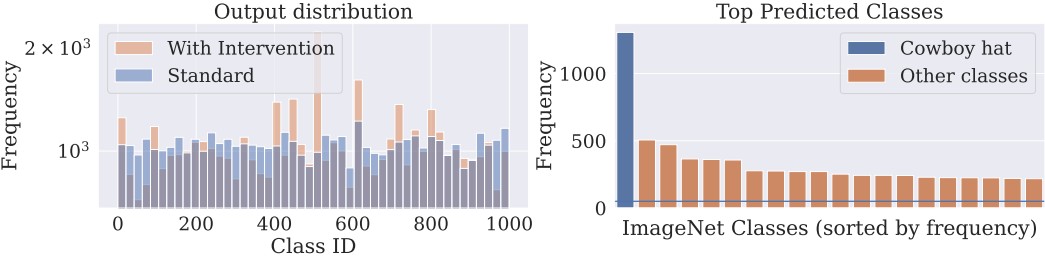

(b) ImageNet distribution shift after intervention.

Figure 24: The **hat** bias in ImageNet.

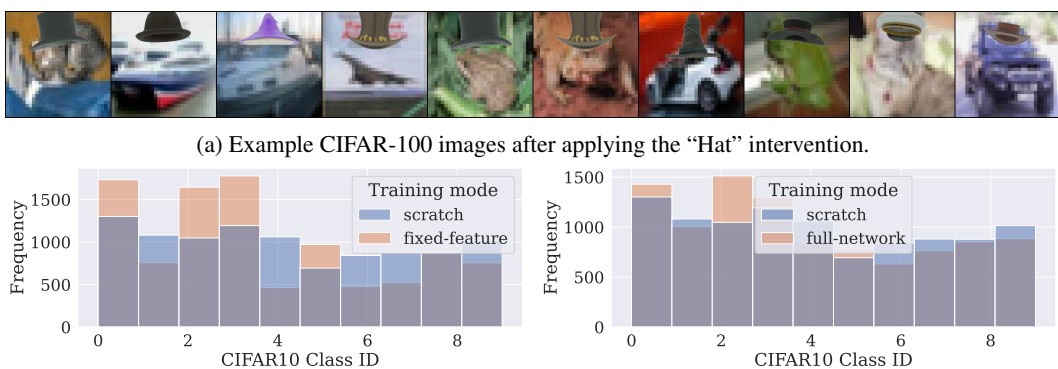

(a) Example CIFAR-100 images after applying the "Hat" intervention.

(b) Output distribution of CIFAR-10 models with the Hat intervention.

Figure 25: The **hat** bias transfers to *CIFAR-10*.

### B.3 Tennis ball bias.

In this section we show the results for the "tennis ball" bias transfer. We first demonstrate in Figure 26 that the "tennis ball" bias actually exists in ImageNet. Then in Figures 27, 28, 29, and 30, we show the output distribution—after applying a tennis ball intervention—of models trained on various datasets either from scratch, or by transferring from the ImageNet model. The from-scratch models are not affected by the tennis ball intervention, while the ones learned via transfer have highly skewed output distributions.

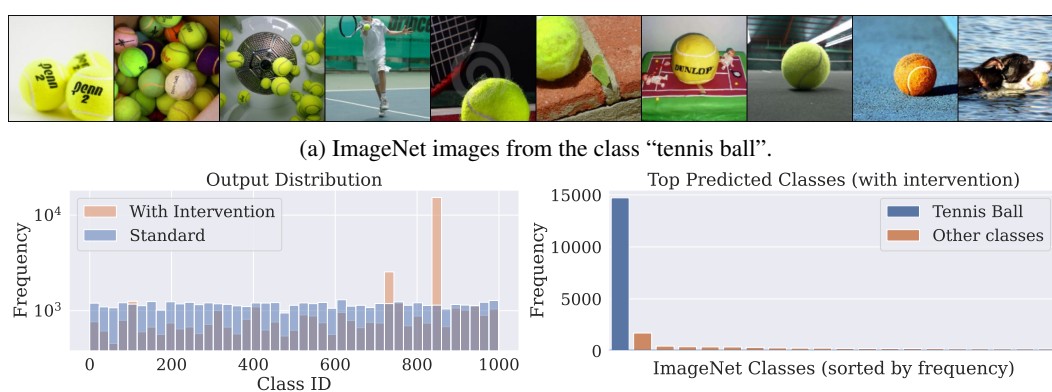

(a) ImageNet images from the class "tennis ball".

(b) ImageNet distribution shift after intervention.

Figure 26: The **tennis ball** bias in ImageNet.

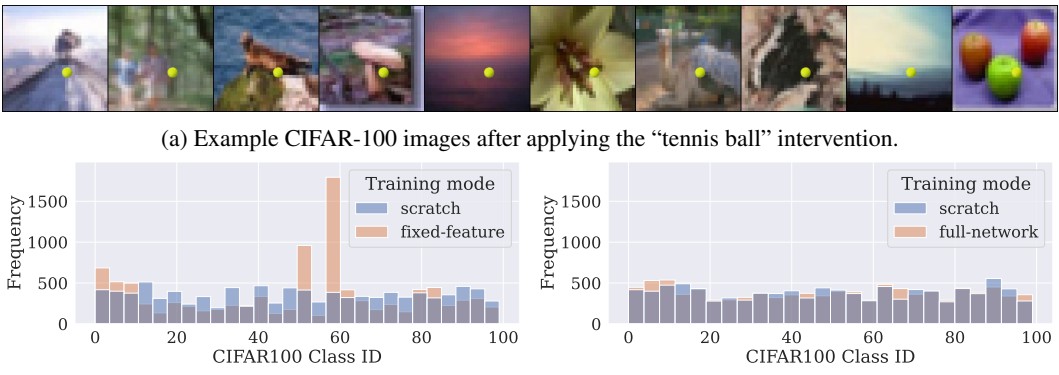

(a) Example CIFAR-100 images after applying the "tennis ball" intervention.

(b) Output distribution of CIFAR-100 models with the tennis ball intervention.

Figure 27: The **tennis ball** bias transfers to *CIFAR-100*.

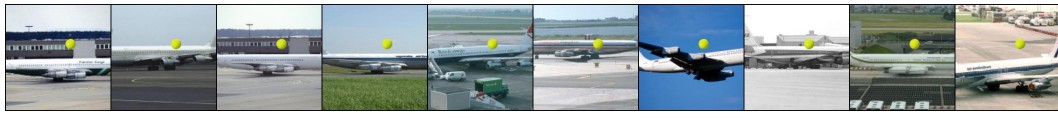

(a) Example Aircraft images after applying the "tennis ball" intervention.

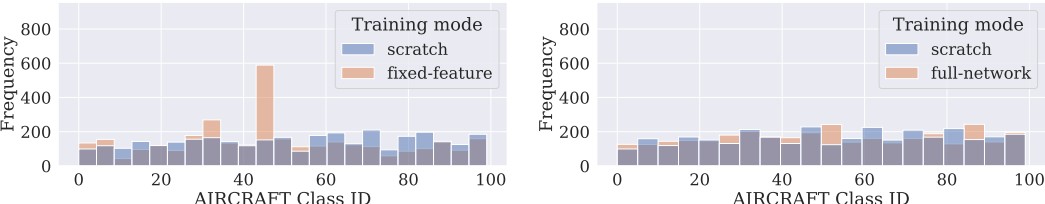

(b) Output distribution of Aircraft models with the tennis ball intervention.

Figure 28: The **tennis ball** bias transfers to *Aircraft*.

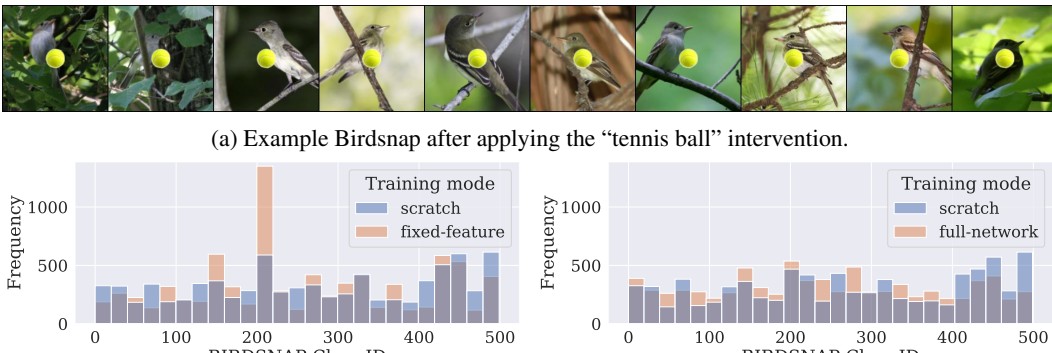

(a) Example Birdsnap after applying the "tennis ball" intervention.

(b) Output distribution of Birdsnap models with the tennis ball intervention.

Figure 29: The **tennis ball** bias transfers to *Birdsnap*.

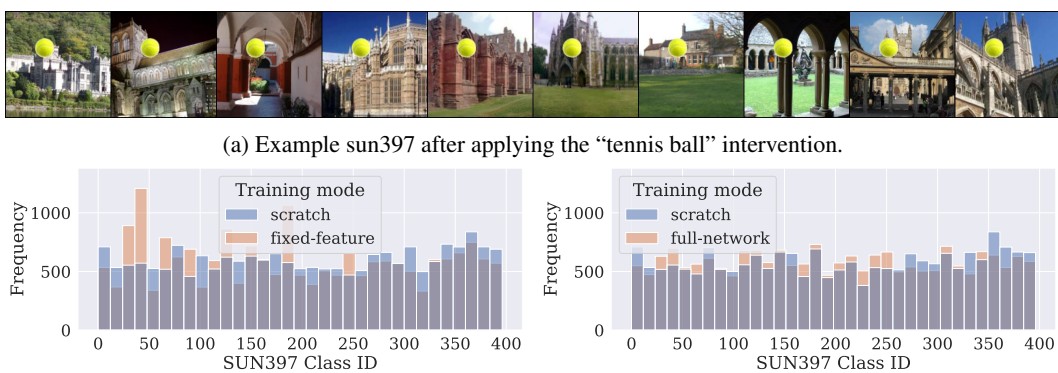

(a) Example sun397 after applying the "tennis ball" intervention.

(b) Output distribution of SUN397 models with the tennis ball intervention.

Figure 30: The **tennis ball** bias transfers to *SUN397*.

