# OpenReview forum: "When Does Bias Transfer in Transfer Learning?"
_ICLR.cc/2024/Conference — Submitted to ICLR 2024_

### Official Review · Reviewer_Fi12 · 2023-10-30

**Soundness:** 3 good
**Presentation:** 3 good
**Contribution:** 1 poor
**Rating:** 5
**Confidence:** 3

**Summary:**

The authors empirically investigate whether biases contained in a pre-trained DNN is transferred to a fine-tuned DNN, in different experimental settings. They confirmed that such biases are actually transferred in (1) synthetic settings using backdoor attacks, (2) synthetic settings with naturally introduced biases of class information (even with de-biased target datasets in fixed-feature setting), and (3) standard transfer learning scenarios on ImageNet.

**Strengths:**

- Their motivation is clear and writing is easy to follow.
- Their experimental scenarios are well-designed. In particular, the phenomenon of transferrability of backdoor attacks is new to me and seems intriguing, but less confident on its novelty since I'm not an expert of ML security.
- Their experiments are thoroughly conducted on vision datasets, and the results are convincing.

**Weaknesses:**

- The novelty and contribution of their findings is limited. Previous works [1][2] already investigated such aspects of transfer learning, and some findings in this submission (particularly the bias transfer phenomenon in the scenario (2) and (3)) can be implied from their results.
- The definition of "bias" in this submission is unclear. It should be specified to discuss "bias" transfer in a possibly rigorous way. Also, I'm less confident whether backdoor attacks should be considerred as "bias", but the research direction of transferrability of such attacks itself should be new and encouraged.
- Discussions on previous works is not enough. The most related works [1][2] are not cited and not discussed. In relation to transferrability of backdoor attacks, I think [3] is one of very related works, but is not discussed. I recommend the authors to survey their previous works and make clear the novelty and contribution of this paper.

[1] B. Neyshabur et al.,  "What is being transferred in transfer learning?" (NeurIPS'20)

[2] E. Lubana et al., "Mechanistic Mode Connectivity" (ICML'23)

[3] A. Shafahi et al., "Adversarially robust transfer learning" (ICLR'20)

**Questions:**

1. What is the definition of biases in this paper? It should be specified first of all to discuss "bias" transfer.

---

> ### Author Response · Authors · 2023-11-18
>
> We thank the reviewer for their response! We discuss their points below.
>
> **W2. Defining Bias:** We define bias as a feature that the model relies on but is not causally linked with the target class. In particular, we focus on biases that the transfer learning model would not have relied on if it had been trained from scratch. We have added this definition to Section 2 in our revision.
>
> **W1, W3. Novelty and Related Work:** Neyshabur et. al focuses on understanding whether low-level statistics or feature reuse is transferred during transfer learning. However, they do not investigate the problem of bias transfer: where features from the pre-training data (that are not used during downstream training) can persist during downstream deployment. Shafahi et al focus on adversarial robustness: in our paper, we focus on biases as features (and in particular, emphasize natural biases which might appear in standard.
>
> Lubana et. al is a recent work (ICML 2023) which performs mechanistic analysis. As part of their analysis, they perform fine-tuning from a backdoored CIFAR-10 to a clean CIFAR-10, and find that fine-tuning may not eliminate spurious cues. In our work, we perform a larger study that fully investigates the landscape of bias transfer across a wide range of tasks for different fine-tuning regimes. We further discuss potential mitigation strategies, such as target dataset debiasing and weight decay. Finally, we investigate bias transfer beyond synthetic biases, and find evidence of such transfer in natural datasets such as ImageNet.
>
> We thank the reviewer for bringing these to our attention and have added them to the related work of the final version.

---

> > ### Comment · Reviewer_Fi12 · 2023-11-21
> >
> > Thank you for the clarification, but some of my concerns still remain.
> >
> > > We have added this definition to Section 2 in our revision.
> >
> > The definition is still highly ambiguous. What does "feature" refer to? What does "a model relies on" mean? How is "not causally linked to the target task" defined? If these are not properly defined, the novelty of this work is also unclear.
> >
> > > Neyshabur et. al focuses on understanding whether low-level statistics or feature reuse is transferred during transfer learning. However, they do not investigate the problem of bias transfer: where features from the pre-training data (that are not used during downstream training) can persist during downstream deployment.
> >
> > I still wonder if "bias transfer" is just a special case of the feture reuse phenomena studied in Neyshabur et al, and thus the novelty of this submission seems to be limited. Nevertheless I'm also less confident in this point because the definition of "feature" in this submission is still ambiguous.

---

> > > ### Author Response · Authors · 2023-11-22
> > >
> > > Thank you for your response. We address your questions below:
> > >
> > > **Definition**: We define bias more formally below (and have included this definition in Appendix A.3 in our revision).
> > >
> > > For input $x \in \mathcal{X}$, let $Y$ be a random variable corresponding to the true label, and let $\hat{Y}$ be a random variable corresponding to the predicted label.
> > >
> > > We can define a "feature" as a transformation  $T: \mathcal{X} \to \mathcal{X}$ which applies some property to an input $x$. For example, if the feature is "black background", the transformation $T$ would apply a black background to $X$. We say that example $x$ "has" a feature $T$ if $x = T(x)$ (adding the feature did not change the input).
> > >
> > > We say that feature $T$ is a $\delta$ ``bias" for a model if:
> > > $$\text{P}(\hat{Y}|T(x)) - \text{P}(\hat{Y}|x) > \delta + \text{P}(Y|T(x)) - \text{P}(Y|x).$$
> > > That is, if the predicted label is more sensitive to the addition of the feature than the true label.
> > >
> > > The phenomenon of bias transfer is when $T$ is a $\delta$-bias for a model pre-trained on a source dataset, but is not a $\delta$-bias for a model trained from scratch (i.e., without pre-training).
> > >
> > > **Feature Re-use**: That transfer-learning models re-use features for downstream tasks is a fundamental aspect of transfer learning (and indeed, this is precisely why transfer learning is so useful). Neyshabur et. al consider feature re-use in the traditional sense (the model fine-tuned on the downstream task uses a feature learned during pre-training). In our paper, we explore a specific vulnerability: we show that if the pre-training model learns a feature which is *not* used during downstream training, the fine-tuned model will still be sensitive to that feature (see the definition of bias transfer above). We find that this sensitivity can cause unexpected downstream deployment failures. Thus, the regime which we consider is different than the feature re-use from Neyshabur et. al

---

> ### Comment · Reviewer_Fi12 · 2023-11-23
>
> Sorry for my late response.
>
> Thank you for clarifying your definition of features and bias transfer. Now I understand that a feature is defined as a transformation in the input domain, which seems a non-standard definition of "feature" though, and the bias transfer is defined counterfactually with such transformation. With these definitions, I also understand that backdoor attacks can be considered as a bias. But I still find it difficult to understand without these definitions which is also only discussed very roughly in Appendix.
>
> > In our paper, we explore a specific vulnerability: we show that if the pre-training model learns a feature which is not used during downstream training, the fine-tuned model will still be sensitive to that feature (see the definition of bias transfer above). We find that this sensitivity can cause unexpected downstream deployment failures.
>
> After reading this explanation, I still consider that it is indeed a special case of feature reuse and thus the findings are naturally implied from the previous results, which itself is ok but leads to the limited novelty and impact of the paper. Also the authors should discuss in more detail about the relationship to the feature reuse, including how it can be suggested from the previous work, rather than just citing the papers in Related Work.

---

### Official Review · Reviewer_HsVn · 2023-10-31

**Soundness:** 3 good
**Presentation:** 3 good
**Contribution:** 2 fair
**Rating:** 3
**Confidence:** 4

**Summary:**

This paper explores the transfer of biases as a result of transfer learning from the source dataset to the transferred models. For both natural and synthetically generated biases, it is shown with experiments that biases pre-existing in the pretraining data get transferred to the downstream tasks, even when the downstream dataset is balanced. The extent of the biases is lesser when finetuning is allowed into the entire network as compared to the case where only retraining the final layer is allowed.

**Strengths:**

1. The study is important, as using pretrained models to finetune on a downstream task is highly beneficial and a popular norm in the current times, hence understanding how the biases in the pretraining datasets creep into the downstream task is necessary to get unbiased predictions.
2. The paper explores multiple settings. They show what happens when the pretraining dataset is biased, where the biases can be both synthetic and natural.
3. The fact that biases are transferred even when the target dataset is debiased is very interesting.
4. Three simple methods have been discussed to reduce the effect of the biases - full network transfer learning, reducing weight decay, mitigate biases in the target dataset.

**Weaknesses:**

1. Novelty is a concern for this paper: all the observations in the paper are expected and not surprising. For example, isnt it obvious that the full network transfer learning setting will be less affected by the source biases than the fixed one?
2. I agree that identification of the problem is certainly important, and this paper does that - the authors demonstrate effectively how dangerous the pretraining data can be in terms of fairness. However, some mitigation strategies or atleast thoughts are expected. One of the solutions proposed is to use full network transfer learning. But if enough resources are not there for a model-user to finetune the entire network, the user has to rely on the fixed feature transfer learning - or settle for something in the model. How to solve the problem in that case?
3. Wang et al [1] suggest manipulating the finetuning data to reduce the biases. No suggestion is proposed by the authors.
4. For the synthetic bias case, what is termed as backdoor attack is simply adding a spurious correlation synthetcially to the dataset to increase/induce bias into it.

[1] Wang et al. 'Overwriting Pretrained Bias with Finetuning Data', ICCV 2023.

**Questions:**

We use pretrained models for a multitude of tasks.
1. What if the pretraining and finetuning data are not entirely similar, and the latter has its own biases? Any suggestions or experiments for such a situation?
2. What happens when the latter is balanced?

---

> ### Author Response · Authors · 2023-11-18
>
> We thank the reviewer for their response, and address their concerns below.
>
> **W1. Novelty:** We want to point out that the main message of our paper is that *spurious features which are present in the pre-training data and completely absent in the downstream task can unexpectedly pop up during downstream deployment*. The rest of our paper (comparing fixed-feature to full-network fine tuning, varying weight decay, looking at different downstream tasks) is all in an effort to fully characterize this phenomenon, but they are not meant to be “surprising” in and of themselves.
>
> **W2. Mitigation Strategies:** In our paper, we discuss several mitigation strategies. In addition to full-network fine-tuning, we find that weight decay and de-biasing can partially mitigate bias transfer (see Section 3.2). Indeed, a major take-away from our paper is that, while mitigation strategies such as weight decay and dataset debiasing can help, such post-hoc strategies typically do not fully remove bias transfer and often are not feasible in many settings (see Reviewer RsQk’s comment). As a result, we believe it is critical to prevent such biases at their source by more carefully choosing our pre-trained models in the first place.
>
> **W3. Manipulating the target dataset:** In our overall framing, we consider the case where the target dataset is balanced (e.g., does not have the bias) but the source dataset is skewed. In this case, bias transfer occurs, even though the bias is not present in the downstream dataset. Furthermore, in Section 3.2 and 4.2, we consider actively debiasing the dataset to counter the bias. However, we find that debiasing the dataset only partially mitigates bias transfer (especially in the fixed transfer case).
>
> **W4. Backdoor attack is just inserting a feature.** We agree that backdoors are just features, which is why we believe it is a useful starting point for studying bias transfer (since biases are also features). We use the standard conception of backdoor attacks found in the data poisoning literature, see e.g., [1, 2].
>
> **Q1. Biases in the downstream dataset:** The scenario where the downstream dataset is biased falls within the standard supervised setting (where the training dataset has biases and struggles on minority groups). Since there is already a vast amount of literature on debiasing in this regime (see for example, Group DRO, Deep Feature Reweighting, Just Train Twice), we do not focus on that regime here.
>
> **Q2. What happens when the latter is balanced?** As we show in Sections 3.2 and 4.2 of our paper, even if the downstream dataset is balanced or unbiased, bias can still transfer! This means that ML developers must pay attention to the pre-training models/data, and cannot rely on solely downstream balancing.
>
> [1] https://arxiv.org/abs/1708.06733
>
> [2] https://arxiv.org/abs/1912.02771

---

> > ### Comment · Reviewer_HsVn · 2023-11-22
> >
> > Thank you for your explanations.
> >
> > W2. Mitigation Strategies: "As a result, we believe it is critical to prevent such biases at their source by more carefully choosing our pre-trained models in the first place."
> > I agree that weight decay etc have been proposed, but following the final suggestion provided by the authors of carefully choosing a pretrained model that does not contain any biases is not trivial. These models are trained on enormous amounts of data (most of the time we may not even know about the pretraining data), and having a model that is completely unbiased is almost impossible; i.e. we have to live with these models and utilize their vast knowledge as well. We have to look for solutions that can debias the downstream predictions even after choosing a biased model.
> >
> > Q1. Biases in the downstream dataset. To clarify, my question was based on what will happen if biases are present in the downstream dataset alongside the pretrained model. What happens when the biases in the pretrained model are similar to that of the downstream data - would there be any amplification? What happens when the biases in the pretrained model and the downstream data are different - can the pretrained model lead to unbiased generations, as it doesnt encode the downstream biases?

---

> ### Author Response · Authors · 2023-11-22
>
> Thank you for your response!
>
> **[W2]**
>
> We agree with the reviewer that no mitigation strategy is fully effective, and that indeed the only foolproof way to avoid bias transfer is to ensure that the pre-training data and model are unbiased. However, we believe that this message (along with the partial mitigation strategies we discuss in our work) only strengthen the message of our paper: **bias transfer is a troubling phenomenon, and has important implications for both deployment and regulation of these models**.
>
>  Our paper thus advocates for further transparency for pre-training models: for example, publishers of such models might list potential biases (similar to Datasheets [1]) when releasing checkpoints. Constructing solutions for detecting and fully countering bias transfer is an exciting avenue for future work (however, our goal in this paper is to show the existence of and characterize bias transfer).
>
> **[Q1]**
>
> We apologize for misunderstanding the reviewer's question here! Since time is rather limited, we ran the following experiment, which we are happy to fully flesh out and include in the final version of the paper. We consider the backdoor experiment from Section 3: where a square is placed on the dog classes of ImageNet, and then either the biased or unbiased ImageNet is fine-tuned on CIFAR-10. This time, however, we consider the case where the downstream dataset *also* has the same spurious feature, by placing a square on the dog class in CIFAR-10 with varying percentages. Our results can be found in Appendix A.9 of the revision: however we summarize them in a table below.  The first row is our standard bias-transfer setup (no bias in the downstream dataset).
>
> *Fixed-feature:*
> |Degree of Fine-Tuning bias| ASR (unbiased source model) | ASR (biased source model)|
> | :--------: | :-------: | :-------: |
> |0% (no bias)  | 6.54| 62.67 |
> |30%   | 13.21| 83.43 |
> |50%  | 18.50| 84.17 |
> |70%  | 21.82| 86.06 |
>
> *Full-network:*
> |Degree of Fine-Tuning bias| ASR (unbiased source model) | ASR (biased source model)|
> | :--------: | :-------: | :-------: |
> |0% (no bias)  | 2.08|  16.49|
> |30% | 23.89 | 40.91|
> |50%  | 35.98| 49.10 |
> |70%  | 44.29| 61.93 |
>
> We find that when the downstream dataset is already biased, pre-training with the biased ImageNet significantly amplifies the downstream bias in both the fixed-feature and full-network settings.
>
> [1] Gebru, T., Morgenstern, J., Vecchione, B., Vaughan, J. W., Wallach, H., Iii, H. D., & Crawford, K. (2021). Datasheets for datasets. Communications of the ACM, 64(12), 86-92.  https://arxiv.org/abs/1803.09010

---

### Official Review · Reviewer_RsQk · 2023-10-31

**Soundness:** 3 good
**Presentation:** 2 fair
**Contribution:** 3 good
**Rating:** 6
**Confidence:** 4

**Summary:**

The authors demonstrate how dataset-induced biases persist after fine-tuning a model, even if the target set does not contain those biases.
For this purpose, the authors designed experiments to introduce or amplify a specific bias and to gauge its presence on the target domain.
The authors explore three mitigation strategies of this bias, including full-network fine-tuning, weight decay, and de-biasing the target domain.

======
Update after rebuttal: I appreciate the additional analysis the authors provided to explain the role of weight decay in mitigating the bias. In its current form the explanation only applies to simple linear regression, and does not extend to a non-linear deep neural network.
Overall, I feel the authors made several points in their analysis which leave the reader with more questions than answers and wishing for more in-depth analysis.
However, given the importance of those points, I am raising my overall score.

**Strengths:**

- Studying bias transfer is important due to the heavy reliance on foundational models.
- The results are insightful and their implications are nontrivial.

**Weaknesses:**

- The work is rather incremental to recent work in the literature, especially the work by Wang and Russakovsky [1]. I missed a reference to that work. The novelty would be more obvious e.g. had the authors demonstrated their results beyond the vision modality. See this recent survey for an overview of closely-related pieces of work, where a proper comparison would help highlight the novelty of the presented work https://arxiv.org/abs/2310.17626
- The mitigations explored seem preliminary or non-straightforward to replicate:
  - Full-network fine-tuning obviously has a better chance of reducing the bias in the pre-trained backbone, compared with a frozen backbone (where the bias mainly exists) + a linear head.
  - The experiments about weight decay do not explain why it is helpful. Is it generally the case that regularization helps mitigate the bias? Is there something specific to weight decay that helps reduce the bias? What about other regularization strategies?
  - Modifying the target dataset to counter the bias seems helpful but it is not obvious how it can be done in the general case (e.g. beyond balancing the sample in different subgroups or reintroducing the backdoor attacks in the target dataset at random).

[1] Wang and Russakovsky: Overwriting Pretrained Bias with Finetuning Data (CVPR '23)




A few typos:
datapoints => data points
can substantially reduces
adjusting [..] entirely eliminate => eliminates
with of people => with people

**Questions:**

- Would adversarial pre-training offer a good mitigation strategy as well?

- The authors mention that they "find that weight decay does not reduce bias transfer in the fixed feature transfer learning
regime, where the weights of the pretrained model are frozen.". How is weight decay applied to frozen weights?

---

> ### Author Response · Authors · 2023-11-18
>
> We thank the reviewer for their response! We address the raised points below.
>
> **W1. Wang et. al:** We thank the reviewers for bringing this to our attention. Wang et. al investigates gender biases in the case where the spurious feature is present in both the pre-training dataset and the fine-tuning dataset. They then consider actively changing the fine-tuning dataset to counter the correlation in the pre-training data (this is equivalent to the “debiasing” scheme in our paper). In our work, we find that (especially in the fixed feature case) a large portion of the dataset would need to be debiased in order to mitigate bias transfer. As you mention, this level of downstream debiasing may not be feasible.
>
> Our work further highlights a different risk: spurious features which are present in the pre-training data and completely absent in the downstream task can unexpectedly pop up during downstream deployment. These types of “backdoor” features are especially pernicious, since a practitioner might not be aware that their pre-training model has such features. We fully investigate the landscape of this phenomenon, across different types of biases (synthetic and natural), architectures, and fine-tuning regimes (fixed-feature vs. full-network). While we discuss potential mitigation strategies (e.g., weight decay), we arrive at a very different conclusion than Wang et. al: practitioners cannot solely depend on post-hoc strategies to deal with bias transfer. As a result, we believe it is critical to prevent such biases at their source by more carefully choosing our pre-trained data in the first place.
>
>
> **W2a. Full-network vs fixed-feature effectiveness is obvious.** We agree with the reviewer that fixed-feature fine-tuning being more prone to bias transfer than full-network fine-tuning is not surprising. Our goal here is to fully investigate the discrepancy between these two fine-tuning strategies, particularly in light of the fact that fixed-feature fine-tuning is becoming increasingly popular in the age of large pre-trained (foundation) models. We thus view quantifying and ablating the gap between the two as a valuable contribution.
>
> More generally, the main finding of our paper is that *spurious features which are present in the pre-training data and completely absent in the downstream task can unexpectedly pop up during downstream deployment*. The remainder of our paper (including the comparison of  fixed-feature to full-network fine tuning) is all in an effort to fully characterize this phenomenon, but is not meant to be “surprising” in and of itself.
>
> **W2b. Weight Decay:** In appendix A.5, we discuss a theoretical justification for why weight decay can help mitigate the bias. Specifically, we consider the logistic regression example from Section 2. The gradient of the logistic loss with L2 regularization becomes
>
>
> $$\nabla \ell_{\mathbf{w}} (\mathbf{x}_i, y_i) = (\sigma(\mathbf{w}^\top \mathbf{x}_i) - y_i)\cdot \mathbf{x}_i + \lambda\mathbf{w}=(\sigma (\mathbf{w}_S^\top \mathbf{x}_i) - y_i)\cdot \mathbf{x}_i + \lambda(\mathbf{w}_S +\mathbf{w}\_{S'})$$
>
> Where $\mathbf{w}\_S$ and $\mathbf{w}\_{S’}$ are the projections of $\mathbf{w}$ onto the span of the target datapoints ($S$) and its complementary subspace ($S’$). The gradient (as in Section 2) restricts the space of updates to those in $S$. However, due to regularization, $\mathbf{w}_{S’}$ is driven to 0, so any planted bias in $S’$ collapses with regularization.
>
> **W2c. Debiasing the target dataset**: We fully agree with the reviewer here—in fact, we believe that (a) according to our paper, debiasing the target dataset does not completely eliminate bias transfer, and (b) “fully” debiasing a dataset is often intractable, as the reviewer notes. This trend holds more generally: while several mitigation strategies can help reduce bias transfer, post-hoc approaches typically do not fully remove it (and often are not infeasible without sacrificing something performance). We thus believe it is critical to prevent such biases at their source by more carefully choosing our pre-trained data in the first place.
>
> **Q1. Adversarial Pre-training**: Adversarial pre-training might be an interesting strategy: indeed, previous work has found that adversarially robust pre-training models do seem to transfer better (see [1]). However, these biases are spurious correlations which (at their core) are just features. It’s not clear that adversarial pre-training will be able to reduce reliance on such features.
>
> **Q2. Weight Decay in the Fixed Feature Regime:**
> In this setting, we apply regularization to the linear layer only.
>
> [1] https://arxiv.org/abs/2007.08489

---

> > ### Comment · Reviewer_RsQk · 2023-11-21
> > **On Weight Decay**
> >
> > Thank you for pointing to your explanation of the effectiveness of weight decay. I appreciate the thought you put into this explanation. I have a few questions:
> >
> > - How did $W$ become $W_S$ in your equation?
> > - Why do you assume that the regularization will drive $W_S'$ to 0? Couldn't it stay constant at any arbitrary (possibly small) value? I do not immediately see a mechanism in which SGD consistently drives $W_S'$ to 0.
> > - Have you been able to quantify $|W_S|$ and   $|W_S'|$ over the course of training?

---

> ### Author Response · Authors · 2023-11-21
>
> Hi, thank you for your response!
>
> - Any $\mathbf{w}$ can be written as $\mathbf{w}\_S + \mathbf{w\}_S'$ where $\mathbf{w}\_S $ is $\mathbf{w}$ projected into the span of the training examples $S$ and $\mathbf{w\}_S'$ is $\mathbf{w}$ projected onto the associated null space $S'$. Then  $$\mathbf{w}^\top x_i = (\mathbf{w}\_S + \mathbf{w}\_{S'})^\top x_i = \mathbf{w}_S^\top x_i$$
> because $\mathbf{w}\_{S'}^\top x_i  = 0$ (since $S'$ is the null space of the span of the training examples).
> - Note that
> $$||\mathbf{w}\_S + \mathbf{w}\_{S'}||_2^2 = \mathbf{w}\_S^\top \mathbf{w}\_S + 2\mathbf{w}\_S^\top \mathbf{w}\_{S'} + \mathbf{w}\_{S'}^\top\mathbf{w}\_{S'} = ||\mathbf{w}\_S||_2^2 + || \mathbf{w}\_{S'}||_2^2$$
> since $\mathbf{w}\_S^\top \mathbf{w}\_{S'}  = 0$. Since $ \mathbf{w}\_{S'}$ does not appear in the logistic loss, any solution $\mathbf{w}$ where the component $||\mathbf{w}\_{S'}||_2^2 > 0$ will be suboptimal. The problem is convex, so SGD will lead to an optimal solution which means $||\mathbf{w}\_{S'}||_2^2 = 0$.
> - We have just added an experiment in the uploaded revision of Appendix A.5 which considers a toy logistic regression example. We show that the component of the initial weight vector projected onto the null space of the training examples (i.e., $||\mathbf{w}\_{S''}||$) degrades to 0 with weight decay (but is unchanged without weight decay). Measuring $||\mathbf{w}\_S||$ and $||\mathbf{w}_{S'}||$ for our neural networks in the main experiments is much more involved (one initial strategy might be to study the SVD of the weights with weight decay). We would be happy to pursue this for the final version.
>
> Please let us know if you have any further questions.

---

### Official Review · Reviewer_2a9y · 2023-11-01

**Soundness:** 4 excellent
**Presentation:** 4 excellent
**Contribution:** 3 good
**Rating:** 8
**Confidence:** 4

**Summary:**

The paper shows empirically that bias in the source distribution can transfer to downstream tasks. The work conducts experiments for backdoor attacks, synthetically controlled biases, and naturally occurring biases. The paper analyzes the effect of various experimental parameters such as weight decay and full network fine-tuning versus frozen features.

**Strengths:**

- The motivation and contributions are clear. Understanding how source datasets affect downstream performance, especially in the context of biases and backdoor attacks is highly relevant given how often pretrained models are used.

- The experiments are extremely thorough, looking at various experimental parameters such as full network fine-tuning versus frozen encoder and the effect of weight decay. Various types of biases are analyzed such as backdoor attacks, natural biases, and synthetically induced biases. The experiments are performed with ImageNet as the source which is a reasonable scale and a common pretraining dataset.

- The figures are illustrative and convey the main takeaways of the experiments.

- The theoretical toy problem is interesting and gives potential intuition for why bias may persist through fine-tuning. It would be nice to see experiments looking at whether over-parametrization affects the amount of bias transfer.

**Weaknesses:**

- It would be useful to know how sensitive these conclusions are to fine-tuning hyper-parameters such as learning rate, momentum, and epochs.

**Questions:**

- Did you do experiments looking at the initial learning rate for fine-tuning and how that affects the amount of bias transfer? I would expect higher learning rates would lead to lower bias transfer.

- Do you think these conclusions would hold for other pretrained models like SimCLR and CLIP?

---

> ### Author Response · Authors · 2023-11-18
> **Response to Reviewer 2a9y**
>
> We thank the reviewer for their detailed review! We address the comments below.
>
> **Impact of learning rate:** We replicated our backdoor experiments in Section 3 with varying learning rates (Appendix A.8 in the revision). Similar to weight decay, we find that increasing the learning rate does not mitigate bias transfer in the fixed-feature setting (ASR actually goes up) but does help in the full-network setting. However, increasing learning rate is a fairly invasive intervention, and often has adverse effects on model performance.
>
> **SimCLR and CLIP:** While we focused on supervised pre-training models in our paper, we would expect biases from SimCLR and CLIP to similarly transfer. Indeed, there has been a large body of work indicating that CLIP contains several concerning biases, especially concerning race and gender [1, 2]. Understanding how these types of biases manifest in fine-tuned models is an interesting avenue for future work.
>
> [1] https://arxiv.org/abs/2108.02818
>
> [2] https://arxiv.org/abs/2210.14562

---

> ### Comment · Reviewer_2a9y · 2023-11-23
> **Response to Authors**
>
> I thank the authors for their response. After reading other reviewer discussions, and more thoroughly reading [1], I have decided to maintain my score. The main concern of other reviewers seems to be lack of recommendations for mitigating biased transfer, and the similarity to Wang et al [1].
>
> While [1] tackles a similar question, I think that this paper provides significant additional insights. This paper conducts experiments across multiple architectures, fine-tuning scenarios, and types of biases whereas [1] is relatively limited along these axes. Two empirical findings I think are significant and differentiate the work from those previous are the following:
>
> - Spurious correlations from the pretraining set can imperceptibly persist in the downstream model even after fine-tuning.
> - Showing that the type of biases that previous works induce artificially, are naturally occurring at scale in ImageNet.
>
> Additionally this work provides some theoretically plausible explanation for how over-parametrization enables biases to persist. I think this could be of interest to others and point to future directions of research. Finally, I don't think the paper should be penalized for not solving the problem of bias. The authors explore a few straight-forward mitigation strategies and show that the problem is not easily solved so researchers pretraining models should be careful.
>
>
> [1] Wang and Russakovsky: Overwriting Pretrained Bias with Finetuning Data.

---

### Author Response · Authors · 2023-11-21
**Discussion Period Ending**

As the discussion period comes to an end (November 22nd), we hope that our responses have addressed the reviewers' questions. If there are remaining concerns, we would be happy to continue the discussion!

---

### Meta-Review · Area_Chair_R7wU · 2023-12-11

**Metareview:**

The paper studies bias transfer while adapting a pretrained model to a target dataset. A key finding of the paper is that spurious features which are present in the pre-training data and completely absent in the downstream task can unexpectedly pop up during downstream deployment. Two of the reviewers remain not convinced about the novelty and contributions of the paper after the author response and continue to have concerns on the paper, in particular about the novelty of the findings over the "feature reuse" work of Neyshabur et al - whether the findings can be naturally implied from the results in there; and about the lack of practical mitigation strategies in the paper. Reviewer HsVn also raised the question about the case when biases in the pretrained model and the downstream data are different which would be an interesting experiment to add to the paper.

To turn this into a stronger paper, authors could consider clarifying how their work is fundamentally different from the work of Neyshabur et al cited by the reviewer, and thinking about some practical mitigation strategies. A thorough discussion of all relevant existing work will also help make the paper stronger.

**Justification For Why Not Higher Score:**

Two reviewers still have concerns about the paper which the author response hasn't addressed satisfactorily.

**Justification For Why Not Lower Score:**

N/A

---

### Decision · Program_Chairs · 2024-01-16

Reject